# *Maf* and *Mafb* control mouse pallial interneuron fate and maturation through neuropsychiatric disease gene regulation

Emily Ling-Lin Pai[1,2], Jin Chen[3,4†], Siavash Fazel Darbandi[1†], Frances S Cho[2,5,6], Jiapei Chen[6,7], Susan Lindtner[1], Julia S Chu[5], Jeanne T Paz[2,5,6,8], Daniel Vogt[9], Mercedes F Paredes[2,5,8], John LR Rubenstein[1,8]*

[1]Department of Psychiatry, University of California San Francisco, San Francisco, United States; [2]Neuroscience Graduate Program, University of California San Francisco, San Francisco, United States; [3]Department of Cellular and Molecular Pharmacology, University of California San Francisco, San Francisco, United States; [4]Howard Hughes Medical Institute, University of California San Francisco, San Francisco, United States; [5]Department of Neurology, University of California San Francisco, San Francisco, United States; [6]Gladstone Institute of Neurological Disease, Gladstone Institutes, San Francisco, United States; [7]Biomedical Sciences Graduate Program, University of California San Francisco, San Francisco, United States; [8]The Kavli Institute for Fundamental Neuroscience, University of California San Francisco, San Francisco, United States; [9]Department of Pediatrics and Human Development, Michigan State University, Grand Rapids, United States

**\*For correspondence:**
John.rubenstein@ucsf.edu

[†]These authors contributed equally to this work

**Abstract** *Maf* (*c-Maf*) and *Mafb* transcription factors (TFs) have compensatory roles in repressing somatostatin (SST[+]) interneuron (IN) production in medial ganglionic eminence (MGE) secondary progenitors in mice. *Maf* and *Mafb* conditional deletion (cDKO) decreases the survival of MGE-derived cortical interneurons (CINs) and changes their physiological properties. Herein, we show that (1) *Mef2c* and *Snap25* are positively regulated by *Maf* and *Mafb* to drive IN morphological maturation; (2) *Maf* and *Mafb* promote *Mef2c* expression which specifies parvalbumin (PV[+]) INs; (3) *Elmo1*, *Igfbp4* and *Mef2c* are candidate markers of immature PV[+] hippocampal INs (HIN). Furthermore, *Maf*/*Mafb* neonatal cDKOs have decreased CINs and increased HINs, that express *Pnoc*, an HIN specific marker. Our findings not only elucidate key gene targets of *Maf* and *Mafb* that control IN development, but also identify for the first time TFs that differentially regulate CIN vs. HIN production.

## Introduction

Pallial interneurons (INs) consist of cortical interneurons (CINs) and hippocampal interneurons (HINs). IN pathologies are hypothesized to underlie symptoms of neurological and neuropsychiatric disorders such as autism spectrum disorder (ASD), epilepsy and schizophrenia. One possible explanation is that IN pathology leads to disrupted circuit inhibition, resulting in circuit hyperexcitability and less efficient information processing (*Lim et al., 2018*; *Rubenstein and Merzenich, 2003*; *Sohal and Rubenstein, 2019*; *Yizhar et al., 2011*). Thus, understanding the genetic control of CIN and HIN development and function can have important clinical ramifications.

CINs are generated in three subpallial progenitor domains: the medial and caudal ganglionic eminences (MGE and CGE), and the preotic area (POA) (*Gelman et al., 2011*; *Lim et al., 2018*). MGE and POA-derived CINs express parvalbumin (PV) or somatostatin (SST), while CGE-derived CINs

express either vasoactive intestinal peptide (VIP), cholecystokinin (CCK) or reelin [but lack SST]. HINs are generated by the same general progenitor pools and share similar molecular properties with CINs. However, so far there are no clear single markers that are known to define developing or mature HINs from CINs (*Pelkey et al., 2017*).

We have recently demonstrated that two *Maf* TFs, *Maf* (*c-Maf*) and *Mafb*, function together in the MGE to control the timing and quantity of SST$^+$ CINs that are generated (*Pai et al., 2019*), schematically summarized in *Schemes 1–3*. Loss of *Mafb* and *Maf* in the *Nkx2.1-Cre* lineage (*Maf* cDKO) leads to the overproduction of SST$^+$ CINs at the expense of PV$^+$ CINs. *Maf* cDKOs also have changes in the quantity and distribution of the total *Nkx2.1-cre* lineage INs: reduced CINs (~30%) and increased HINs (~2 fold) at P0, and a progressive loss of INs that plateaued by adulthood at ~70% reduction (*Schemes 1* and *2*). Loss of *Maf* and *Mafb* also alters postnatal CIN properties, including process morphogenesis, synaptogenesis, intrinsic electrophysiology and circuit excitability (*Pai et al., 2019*; *Schemes 2* and *3*). However, the gene targets of *Mafb* and *Maf* that underlie these phenotypes remain to be elucidated.

Herein, we uncovered a set of *Maf* and *Mafb* regulated genes by performing single cell RNA-sequencing (scRNA-seq) of neonatal CINs and HINs from wild type (WT) and cDKO. These include *Sst, neuropilin1* (*Nrp1*), and several genes that are implicated in neuropsychiatric disorders, including myocyte enhancer factor 2C (*Mef2c*) (ASD, *Tu et al., 2017*), synaptosome associated protein 25 (*Snap25*) (schizophrenia, *Houenou et al., 2017*), and C-X-C motif chemokine receptor 4 (*Cxcr4*) (22q11.2 deletion syndrome; *López-Bendito et al., 2008*). We provide evidence that *Maf* and *Mafb* promote *Mef2c* expression, which in turn promotes PV$^+$ IN differentiation. We show that *Maf* and *Mafb* also have postnatal functions in CIN morphological maturation. Furthermore, we identified a

**Scheme 1.** Schema depicting the prenatal roles of *Maf* and *Mafb*. In *Pai et al., 2019*, we described that the expression of *Maf* and *Mafb* starts in the MGE SVZ. Deletion of *Maf* and *Mafb* using the *Nkx2.1-Cre* (*Maf* cDKO) leads to increased neurogenesis in the MGE SVZ and overproduction of somatostatin (SST) interneurons. In this paper, we aim to identify the downstream gene targets of *Maf* and *Mafb* that are involved in this process. SVZ: subventricular zone. MZ: mantle zone.

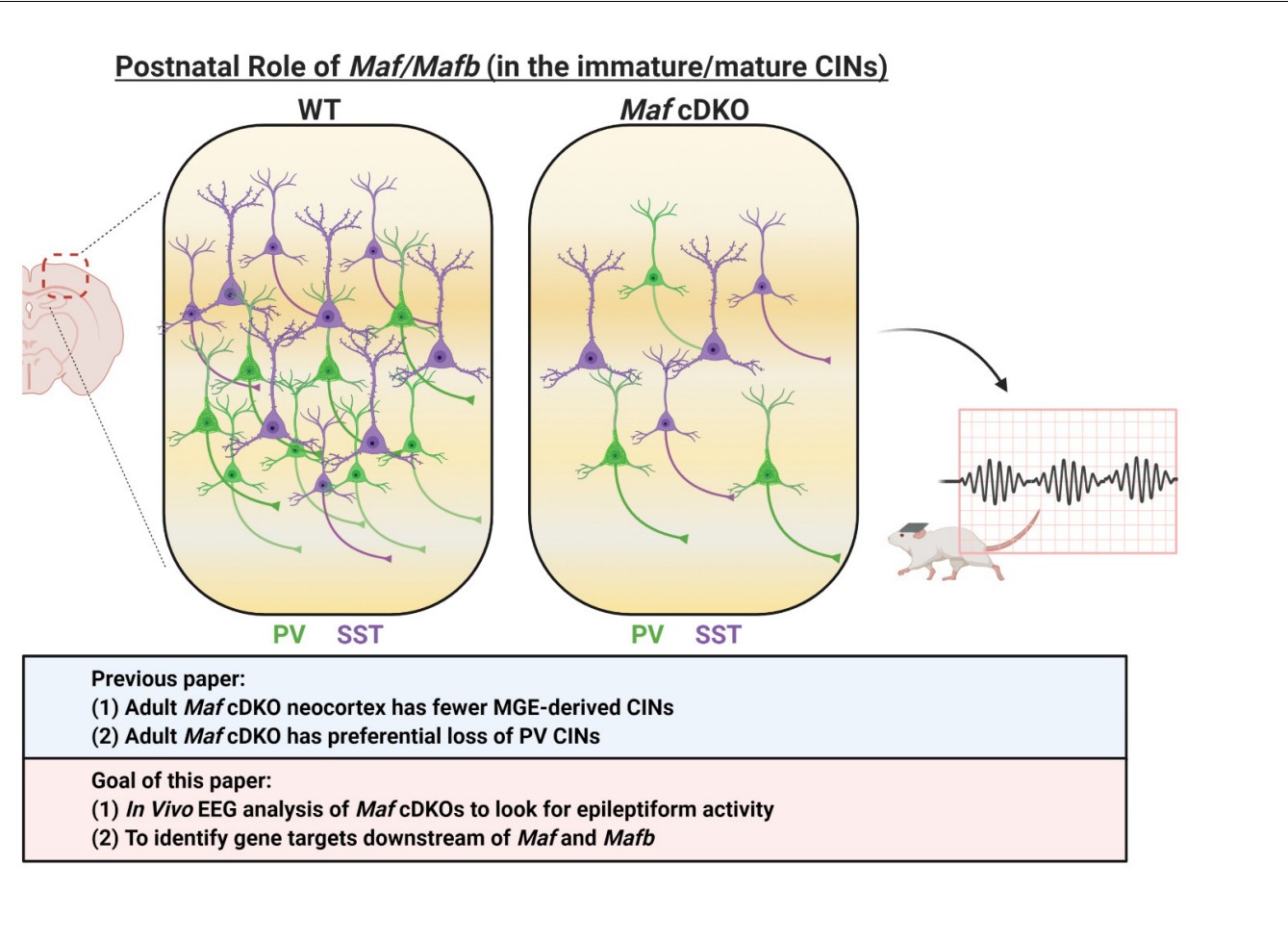

**Scheme 2.** Schema depicting the postnatal roles of *Maf* and *Mafb*. In *Pai et al., 2019*, we discovered that the adult *Maf* cDKO mice have drastic reduction of MGE-derived cortical interneurons (CINs) and a preferential loss of parvalbumin (PV) INs. In this paper, we aim to identify potential downstream gene targets of *Maf* and *Mafb* that are involved in PV CIN fate specification and differentiation. In this paper, we also provide data that suggest adult *Maf* cDKO animals have spontaneous epileptic activity in vivo.

secreted peptide encoding gene, prepronociceptin (*Pnoc*), as an immature HIN marker. Finally, we provide evidence that *Maf* and *Mafb* control the balance between CIN and HIN production.

## Results

### Single cell transcriptomic profiling of P0 WT and *Maf* cDKO

Deletion of both *Maf* (*c-Maf*) and *Mafb* using *Nkx2.1-Cre* (*Maf* cDKO) led to an overproduction of $Sst^+$ CINs, a drastic loss of total MGE-derived CINs and abnormal electrophysiological properties (*Pai et al., 2019*). Furthermore, adult *Maf* cDKOs have spontaneous non-motor epileptic activity, likely associated with the interneuronopathy (*Figure 1—figure supplement 1*). The underlying gene targets of *Maf* and *Mafb* that may regulate these processes are explored below.

To identify dysregulated genetic targets in cDKO immature INs, we used scRNA-seq from P0 WT and cDKO neocortices (*Figure 1A–D*). We pooled the WT and cDKO datasets to perform unsupervised data analysis using Seurat pipeline (*Butler et al., 2018*; *Ge et al., 2020*). We identified 24 different clusters, including excitatory neurons, neural progenitors, microglia as well as MGE and CGE-derived INs, which were visualized by uniform manifold approximation and projection (UMAP) (*Figure 1B and C*; *Becht et al., 2019*). We assigned the cell identities to each cluster based on the

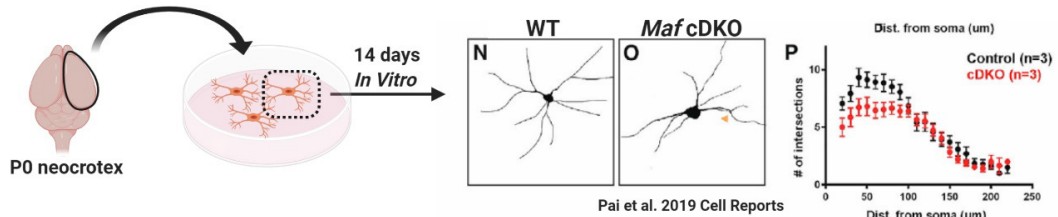

**Scheme 3.** Loss of *Maf* and *Mafb* leads to MGE-derived IN morphogenesis defect. In *Pai et al., 2019*, we showed that cultured neonatal *Maf* cDKO interneurons have neurite outgrowth defect compared to the WT counterparts. In this paper, we aim to identify gene targets of *Maf* and *Mafb* that are involved in CIN morphological maturation.

expression of established marker genes (*Figure 1—figure supplements 2* and *3*, *Figure 1—source data 1*; *Loo et al., 2019*; *Mi et al., 2018*; *Tasic et al., 2018*). MGE-derived INs were in cluster 1, based on marker gene co-expression, including *LIM homeobox 6* (*Lhx6*), *Maf*, *Mafb* and *neurexophilin 1* (*Nxph1*) (*Figure 1E*, *Figure 1—figure supplement 2*, *Figure 1—source data 1*). We performed fluorescent in situ hybridization (FISH) to confirm the specificity of *Nxph1* expression within the *Nkx2.1*-lineage by co-labeling *Nxph1* and *Nkx2.1-Cre* driven *tdTomato* (*Figure 1—figure supplement 4*).

There were no obvious transcriptomic changes in neural progenitors, excitatory neurons, oligodendrocytes, microglia and the CGE-derived IN population between WT and cDKO (*Figure 1B and C*). This suggests no gross non-cell autonomous effects exerted by loss of *Maf* and *Mafb* in the *Nkx2.1-Cre*-lineage. However, the MGE-derived IN population (cluster 1) showed clear differences between WT and cDKO (*Figure 1C*). We performed differential gene expression (DEX) analysis in this MGE-derived IN cluster, and identified 81 differentially expressed (DE) genes, with 47 upregulated (including *Sst*, *Nrp1*, *Pnoc* and *Neuropeptide Y*) and 34 down-regulated (including *Maf*, *Mafb*, *Neurexophilin2*, *Mef2c*, *Cxcr4* and *Snap25*) in the *Maf* cDKO (*Figure 1—figure supplement 5*). We performed either immunohistochemistry or FISH to confirm the pattern of gene expression changes between control and cDKOs, including *Mef2c*, *Sst*, *Neurexophilin2* (*Nxph2*), *Cxcr4*, *Neuropeptide Y* (*Npy*) and *Nrp1* (*Figure 1F–N*, *Figure 1—figure supplements 4–7*). Gene ontology analysis, based upon cellular components, showed that most dysregulated genes are involved in synapse formation, neurite projection and neuronal maturation. Thus, the transcriptomic analysis provides molecular insights into the *Maf* cDKOs cellular phenotypes (*Figure 1—figure supplement 5*; *Ge et al., 2020*; *Pai et al., 2019*).

### Loss of *Maf* and *Mafb* leads to decreased MEF2C expression and increased *Sst*⁺ CINs

Based on DEX analysis, *Mef2c* was the third most down-regulated gene, while *Sst* was the most upregulated gene (*Figure 1F–G*, S7). To validate the *Mef2c* result, we used immunofluorescence labeling; MEF2C expression was reduced in the tdTomato⁺ CINs across all cortical layers in the P2 cDKOs (*Figure 1F and L–N*) (Bin 3, 4, 7: p<0.05; Bin 5, 6: p<0.01; Density: p<0.01; Welch's t test) (tdTomato fluorophore was generated by crossing Ai14 Rosa26-tdTomato *Cre* reporter *with Nkx2.1-Cre*; we will refer to the *Cre* reporter allele as RosaT and the expressed reporter as tdTomato throughout the paper)(*Madisen et al., 2010*). To validate the *Sst* result, we used FISH; we found

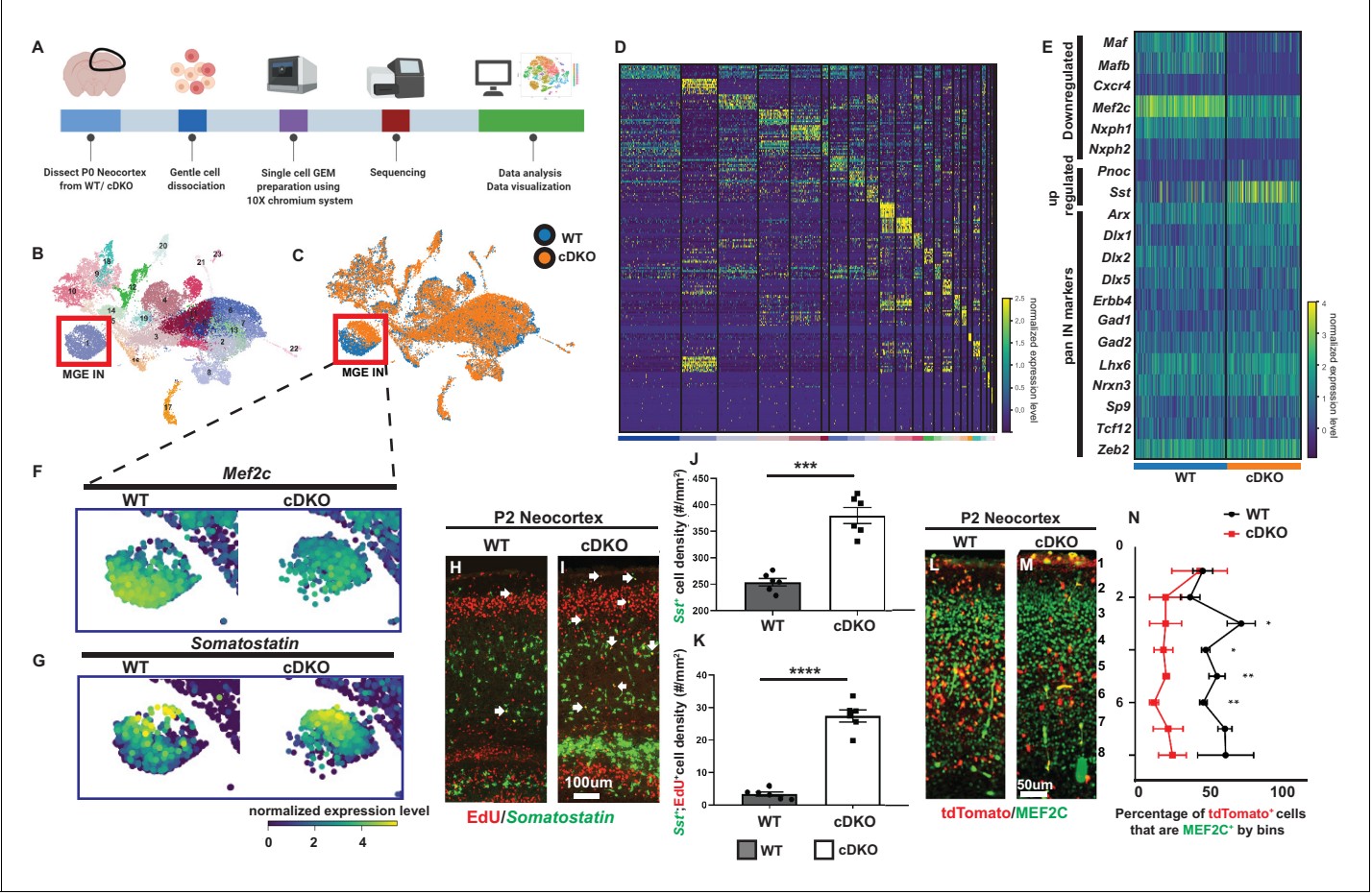

**Figure 1.** Single cell transcriptomic analysis of P0 WT and *Maf* cDKO neocortex. (A) Experimental design for 10X genomics analysis of the P0 WT and *Maf* cDKO neocortex. Neocortical tissues were dissociated for single cell gel bead-in-emulsion (GEMs) particle preparation, followed by cDNA library preparation, sequencing and data analysis. (B) UMAP plot of WT and cDKO cells, color coded with cluster identities (see *Figure 1—figure supplement 2*). (C) UMAP plot of WT and cDKO cells, color coded with genotypes; cluster one are MGE INs. Note the genotype separation of the cells in cluster 1 (left bottom), but not other clusters, suggesting the lack of non-cell autonomous effects of the *cDKO*. (D) Heatmap showing cluster identities with corresponding marker genes. (E) Heatmap of cluster 1 cells from WT and cDKO showing average expression of selected MGE IN markers and pan-IN markers. Note the nearly full depletion of *Mafb* and *Maf* and increased expression of *Sst*, and the decreased expression of *Mef2c*, while other MGE-derived IN markers such as *Dlx1/2/5*, *Gad2* and *Lhx6* were not changed. This suggests that the deletion of *Maf*s in the MGE-lineage does not lead to a gross change in MGE-derived IN fate. (F–G) Enlarged feature plots showing the expression of *Mef2c* (F) and *Sst* (G) from WT and cDKO from cluster 1. (H–I) FISH in *Sst* together with EdU labeling showing the amount of E15.5 born MGE-derived CINs that became *Sst*+ in WTs and cDKOs. Arrows point to the cells that are EdU+;*Sst*+. (J) Quantification of *Sst*+ CINs at P2 WT and cDKO neocortex. (K) Quantification of EdU+;*Sst*+ CINs at P2 WT and cDKO neocortex. (L–M) Immunofluorescent staining of MEF2C colocalized with *Nkx2.1-cre*-driven tdTomato reporter. P2 *Maf* cDKOs have decreased MEF2C expression (M) compared to WTs (L). (N) Quantification of the proportion of tdTomato+ CINs that were MEF2C+ by bins. N = 3–4 animals per group and multiple brain sections were used for quantification. Scale bar in (I) = 100 um and in (M) = 50 um. *p<0.05, **p<0.01, ***p<0.001, ****p<0.0001. The online version of this article includes the following source data and figure supplement(s) for figure 1:

**Source data 1.** Cluster marker gene list Here we provide a list of genes that were used to define each cluster's identity.

**Figure supplement 1.** Adult *Maf* cDKO animals have spontaneous non-motor seizure activities.

**Figure supplement 2.** Top 10 ranked marker genes for each cluster.

**Figure supplement 3.** Dotplots of non-IN and IN population by selected marker genes.

**Figure supplement 4.** *Nxph1-tdTomato* double FISH and *Nxph2* ISH.

**Figure supplement 5.** DEX gene list and Gene Ontology analysis on DEX genes Heatmap showing the DEX gene list of the MGE IN population (cluster 1) when compared cDKO to WT.

**Figure supplement 6.** Downregulation of *Cxcr4* expression at E15.5 and P2 in the *Maf* cDKO.

**Figure supplement 7.** *Maf* cDKO has increased *Sst*+/*Npy*+/*Nrp1*+ expressing HINs.

both increased *Sst* ISH signal and increased numbers of *Sst*⁺ CINs at P2 in cDKOs as before (*Figure 1G and H–J*) (*Sst* density: p=0.0001, Welch's t test) (*Pai et al., 2019*).

Since the *Maf* cDKO generated excessive *Sst*⁺ CINs at E13.5 (*Pai et al., 2019*), we also asked if excessive *Sst*⁺ CINs were generated at later developmental time points. Thus, we injected EdU in E15.5 pregnant mice and assessed the neocortical *Sst*⁺ INs at P2. There was a > 3 fold increase in *Sst* and EdU double labeled CINs in P2 cDKO neocortices (*Figure 1K*) (*Sst*-EdU density: p<0.0001; Welch's t test). Thus, *Maf* cDKOs overproduce *Sst*⁺ CINs across development.

## *Maf* and *Mafb* drive *Mef2c* expression to promote PV⁺ CIN generation

Prenatal loss of *Mef2c* reduces the number of PV⁺ CINs (*Mayer et al., 2018*). Given our finding that *Mef2c* expression is reduced in *Maf* cDKOs (*Figure 1*), it suggests that *Maf* and *Mafb* are genetically upstream of *Mef2c*. We thus hypothesized that deficits in *Mef2c* potentially underlie the mechanisms causal to the PV⁺ CIN loss in the *Maf* cDKO (this report; *Pai et al., 2019*). We tested this by increasing MEF2C levels in *Maf* cDKO MGE cells.

To this end, we used an MGE transplantation rescue assay (*Vogt et al., 2015*; *Vogt et al., 2017*), which previously showed that the *Maf* cDKO had reduced PV⁺ CINs, recapitulating the cDKO in vivo phenotype (*Pai et al., 2019*). E13.5 MGEs (RosaT⁺ or *Maf^{f/f}*; *Mafb^{f/f}*; RosaT⁺) were dissociated and then transfected with a *Cre*-expressing plasmid to generate tdTomato⁺ WT or cDKO MGE-lineage cells. In parallel, some of the MGE cells were co-transfected with plasmids expressing *Cre* and *Mef2c*. The transfected cells were transplanted into WT P1 neocortices and were assessed 30 days later via SST and PV immunofluorescence (*Figure 2A and C–K*).

The analysis confirmed the preferential loss of PV⁺ CINs in the cDKOs (*Pai et al., 2019*; *Figure 2B*). Importantly, we found that *Mef2c* transfection rescued this phenotype, and corrected the SST/PV ratio (*Figure 2B and K*) (PV population: WT vs. cDKO p<0.0001; cDKO vs. cDKO +*Mef2c* p<0.0001; *Chi*-square test). Together, these data suggest that *Maf* and *Mafb* promote *Mef2c* expression, which in turn promotes the PV⁺ IN fate (*Figure 2—figure supplement 1*. Model).

## Identification of transcriptomic features of candidate immature PV⁺ INs

Studies have strived to identify transcriptional machinery that differentially controls PV vs. SST IN neurogenesis. Our current study has solidified a transcriptional cascade, *Maf/Mafb* → *Mef2c* → PV, that is involved in this process (*Figure 2*). However, there are no known early markers that specifically delineate immature PV⁺ INs before PV expression begins (*del Rio et al., 1994*).

We hypothesize that within MGE-derived INs, the *Sst*-negative (*Sst*⁻) group primarily comprises the future PV⁺ INs, in part since *Sst-IRES-Cre* lineage tracing showed that the majority of the SST⁺ INs remain SST⁺ (*Pai et al., 2019*; *Figure 3—figure supplement 1*). We computationally separated WT cluster 1 MGE-derived INs into *Sst*⁺ and *Sst*⁻ subgroups and evaluated the mean expression of MGE IN markers (*Figure 3A*). We identified a group of genes that are enriched in the *Sst*⁻ population, by comparing the IN marker expression between the two groups (*Figure 3*, *Figure 3—figure supplement 2*). From these, we selected six genes, ADP ribosylation factor like GTPase 4D (*Arl4d*), engulfment and cell motility protein 1 (*Elmo1*), insulin like growth factor binding protein 4 (*Igfbp4*), *Mef2c*, *Sp9* and transcription factor 12 (*Tcf12*), with high-to-moderate expression to evaluate their RNA expression.

We used P2 WT tissues which were triple-labeled with *Nkx2.1-Cre*-driven *tdTomato*, *Sst* and an RNA of each of the six genes; the RNAs were detected using RNAscope (*Figure 3B–G'''*, *Figure 3—figure supplements 2* and *3*). We quantified the proportion of *tdTomato*⁺; *Sst*⁺ or *tdTomato*⁺; *Sst*⁻ INs that were positive for each gene (*Figure 3I, K and M*, *Figure 3—figure supplement 2*). Our results suggested that *Mef2c* is a very specific *Sst*⁻ population marker, especially in the hippocampus (Cortex: p<0.05; Hippocampus: p<0.0001). Likewise, *Elmo1*, *Igfbp4*, and *Sp9* appear to be good markers for *Sst*⁻ MGE-derived INs in the hippocampus (*Elmo1*: p<0.001; *Igfbp4*: p<0.05; *Tcf12*: p<0.01; unpaired test). This likely is the first study to identify genes, such as *Mef2c*, that are almost exclusively expressed in *Sst*⁻ MGE-derived immature HINs.

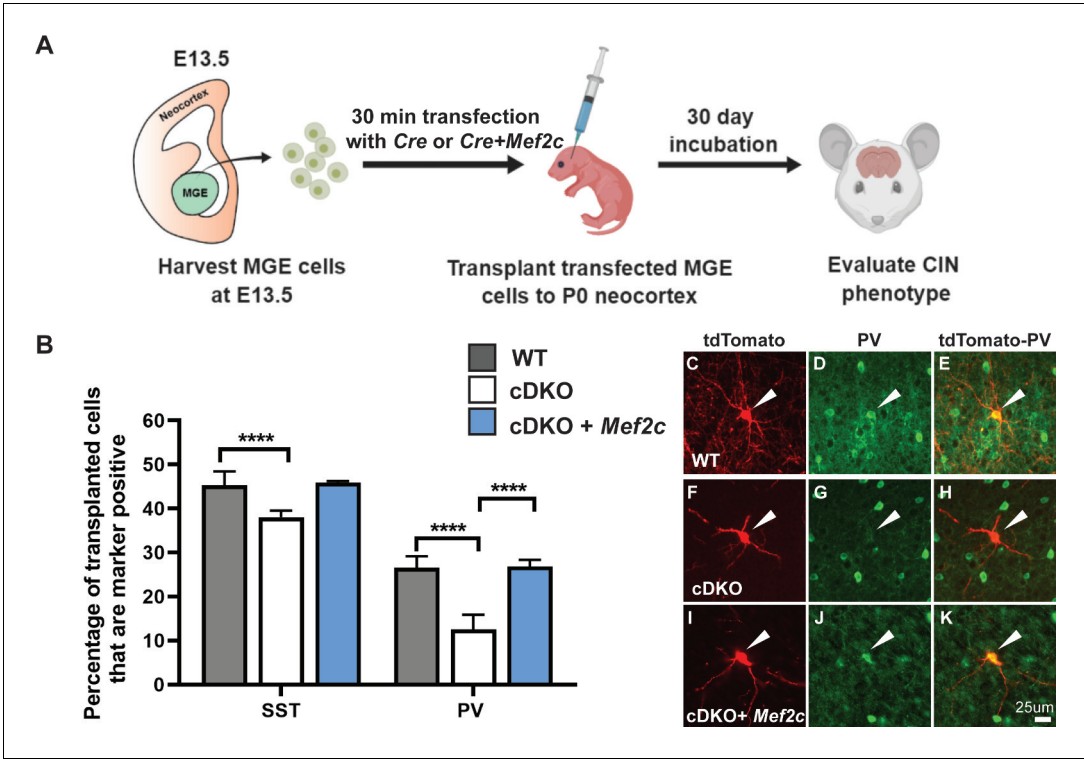

**Figure 2.** *Mef2c* rescues the deficit of PV CIN production in cDKOs. (**A**) Schema showing the MGE transplantation assay. RosaT⁺ or *Maf^{f/f}*;*Mafb^{f/f}*;RosaT⁺ MGE progenitors were dissected at E13.5, followed by transfection of *Cre* or *Cre + Mef2c* (expression vectors) to generate cDKO or cDKO + *Mef2c* MGE cells. We transplanted these cells into P1 WT neocortex and let them mature for 30 days before investigating CIN subtype markers. (**B**) Quantification of the proportion of transplanted WT, cDKO or cDKO with transfected *Mef2c* CINs that express either SST or PV. Note that the cDKO group recapitulated the phenotype of preferential loss of PV CINs; transfection with *Mef2c* rescued the production of PV CINs. *Chi*-square test was used for statistical analysis. (**C–K**) Representative singular cell images for group WT, cDKO and cDKO+*Mef2c* that are labeled with tdTomato and PV. Note that cDKO (**F**) has less complex neurite growing pattern compared to the WT (**C**). Also note the rescue of the PV expression in the cDKO+*Mef2c* (**K**) compared with the cDKO (**H**). Scale bar in (**K**) = 25 um. For SST IN comparison: WT-701 cells, cDKO-627 cells, cDKO+*Mef2c*- 304 cells; For PV IN comparison: WT- 670 cells, cDKO-587 cells, cDKO+*Mef2c* −236 cells. These numbers came from 6 WT animals, 4 cDKO animals and 3 cDKO + *Mef2c* animals. Cells analyzed were converted to contingency table to conduct *Chi*-square test. ****p<0.0001. Error bars represent standard errors of the mean.

The online version of this article includes the following figure supplement(s) for figure 2:

**Figure supplement 1.** Model.

## *Maf* and *Mafb* control CIN morphological maturation by regulating *Mef2c* and *Snap25*

Reduced expression of *Mef2c* and *Snap25* was shown to impede neurite morphogenesis (*Harrington et al., 2016*; *Houenou et al., 2017*; *Tu et al., 2017*). Since *Maf* cDKOs have a neurite outgrowth defect in vitro and in vivo (*Pai et al., 2019*, *Figure 4—figure supplement 1*), we examined whether the decreased expression of *Mef2c* and *Snap25* in the cDKOs underlies their neurite outgrowth defect (*Pai et al., 2019*).

To evaluate this, we tested whether transfection of these genes could rescue the *Maf* cDKO neurite phenotype using primary cell cultures of P1 neocortical neurons. Expression vectors encoding G*fp*, *Gfp*-tagged *Maf*, *Mef2c* or *Snap25* were transfected into *Nkx2.1-Cre*-lineage CINs (tdTomato⁺) the day after cell plating (DIV1). At DIV 14, CIN morphology was assessed (*Figure 4A*). As before, cDKO CINs had decreased numbers of neurites; *Gfp* transfection did not alter this phenotype (*Figure 4B, C and G*). Second, transfection of cDKO with *Maf*, *Mef2c* or *Snap25* rescued neurite numbers, and the increased neurite numbers were comparable to WT (*Figure 4B, D–F and H–I*).

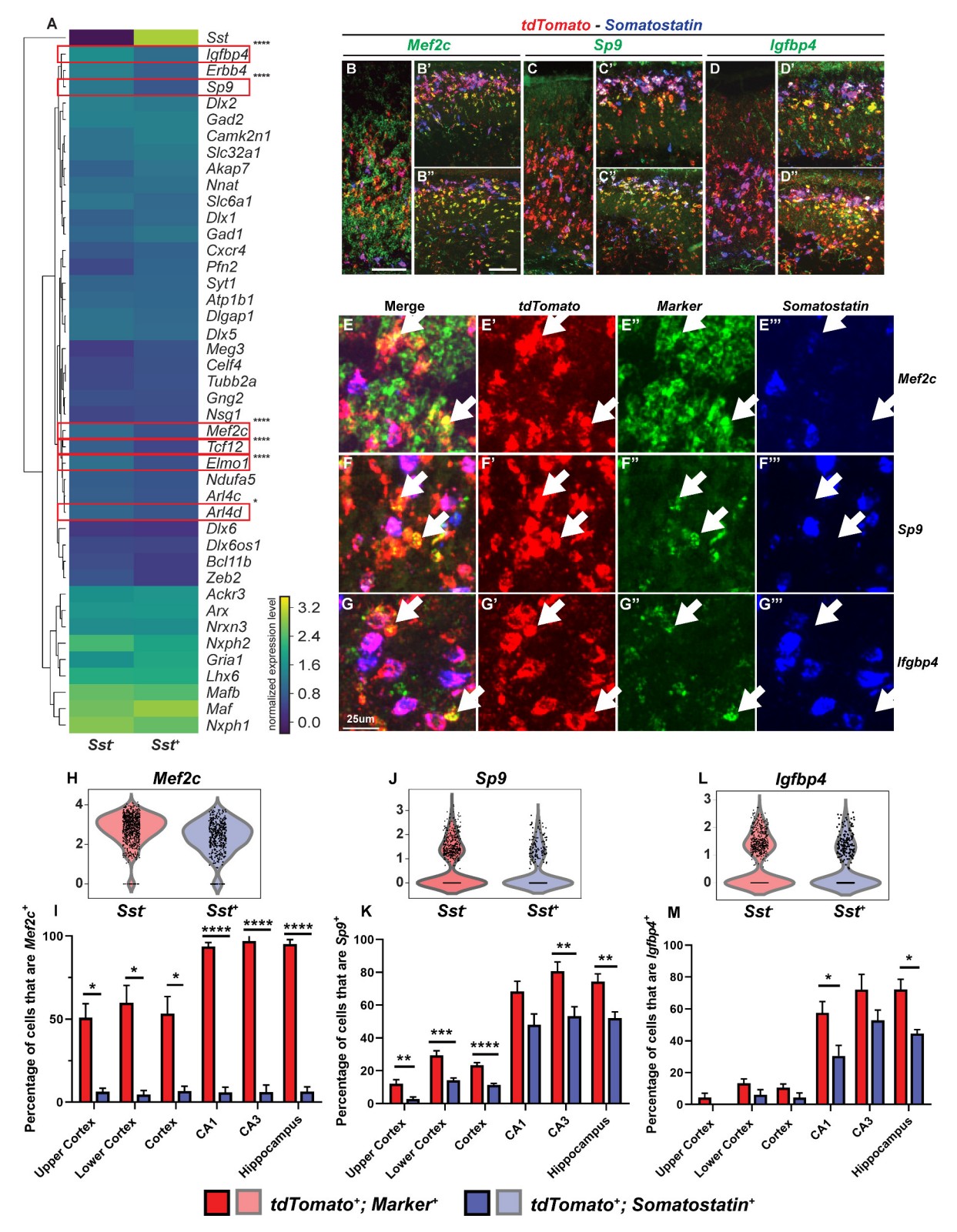

**Figure 3.** Multiplex RNA in situ hybridization validation of proposed immature PV IN markers. (**A**) Heatmap of MGE IN marker genes for cluster 1 *Sst*⁻ and *Sst*⁺ INs. (**B–D''**) Lower magnification fluorescent images of multiplex RNA in situ hybridization for *Mef2c* (**B–B''**), *Sp9* (**C–C''**) and *Igfbp4* (**D–D''**). (**E–G'''**) Higher magnification fluorescent images of multiplex RNA in situ hybridization for *Mef2c* (**E–E'''**), *Sp9* (**F–F'''**) and *Igfbp4* (**G–G'''**). White arrows point to INs that are marker (*Mef2c*, *Sp9* or *Igfbp4*) and *tdTomato* positive but *Sst* negative. Violin plots showing the normalized expression value (Y-

*Figure 3 continued on next page*

Figure 3 continued

axis) of each cell analyzed in each group for *Mef2c* (H), *Sp9* (J) and *Igfbp4* (L). Quantification of the percentage of tdTomato$^+$; Sst$^-$ and tdTomato$^+$; Sst$^+$ INs in the neocortex and in the hippocampus that are either *Mef2c*$^+$(I), *Sp9*$^+$(K) or *Igfbp4*$^+$(M). Scale bar in (B) = 100 um, (B'') = 200 um and (G) = 25 um. 2 WTs and multiple brain sections per animal were used for quantification. For statistical analysis, multiple independent t-tests without same standard deviation assumption were conducted to compare the expression of each gene in each brain region. *p<0.05, **p<0.01, ***p<0.001, ****p<0.0001. The online version of this article includes the following figure supplement(s) for figure 3:

**Figure supplement 1.** Schema depicting the hypothesis behind candidate PV IN marker discovery In *Pai et al., 2019*, we fate-mapped the *Sst-IRES-Cres* lineage and discovered that > 90% of *Sst-IRES-Cre* labeled CINs remain to be SST-expressing in adulthood.
**Figure supplement 2.** Multiplex RNA in situ hybridization validation of proposed immature PV IN markers.
**Figure supplement 3.** Higher magnification view of HINs from multiplex fluorescent in situ hybridization.

These results provide evidence that reduced *Mef2c* and *Snap25* expression are downstream of *Maf* and *Mafb* in regulating neurite complexity in neonatal CINs.

In these experiments, the *Maf* and *Mafb* genes were deleted in the MGE at E10.5 using *Nkx2-1-Cre*. Thus, the CIN phenotypes could be the result of the continuous lack of *Maf* and *Mafb* throughout development and may not necessarily reflect *Maf* and *Mafb* post-mitotic functions. Therefore, we analyzed neuronal morphology in CINs generated from *Maf* cDKOs using *Sst-IRES-Cre*, a *Cre* line that turns on after IN cell cycle exit (*Figure 4—figure supplement 2*). We found that the *Sst-IRES-Cre* generated *Maf cDKO* had neurite defects similar to those found using *Nkx2.1-Cre*. Thus, *Maf* function in neurite outgrowth is also required post-mitotically.

To identify a postnatal function for *Maf* and *Mafb*, we deleted *both* in neonatal CINs using an in vitro primary culture assay. WT and *Mafb*$^{f/f}$; *Maf*$^{f/f}$; RosaT$^+$ cortical neurons were transfected at DIV1 with a *Cre*-expressing plasmid (driven by the *Dlxi12b* enhancer which is specifically active in forebrain GABAergic neurons). CRE mediated recombination generates tdTomato$^+$ cDKOs CINs (cDKO-POST (POSTNATAL)) (*Figure 4J*). At DIV14, we performed Sholl analysis, which revealed that cDKO$^{POST}$ INs had reduced neurite complexity, providing evidence that *Maf* and *Mafb* regulate IN morphology postnatally. We then tested whether *Maf*, *Mef2c* and *Snap25* could rescue this phenotype, by co-transfecting *Maf*$^{f/f}$; *Mafb*$^{f/f}$; RosaT$^+$ cells at DIV1 with plasmids expressing *Cre* and either *Maf*, *Mef2c* or *Snap25*. We analyzed the neuronal morphology at DIV14. *Maf*, *Mef2c* and *Snap25* each rescued the neurite outgrowth defect (*Figure 4K–R*). Thus, *Maf* and *Mafb* likely promote CIN neurite morphogenesis at postnatal stages through *Mef2c* and *Snap25*.

## Prepronociceptin is a HIN marker

The DEX analysis revealed that *prepronociceptin* (*Pnoc*) was an upregulated gene in the *Maf* cDKO (*Figure 1—figure supplement 5*). A recent study focusing on adult CA1 HIN profiling suggested *Pnoc* is a marker of O-LM INs, a subtype that is mainly Sst$^+$ (*Harris et al., 2018*). Thus, we investigated *Pnoc* expression by FISH in the neonatal hippocampus and cortex (*Figure 5A–H*). In P2 WTs, *Pnoc*$^+$ cells were largely restricted to the hippocampus (*Figure 5G*).~40% of the *Pnoc*$^+$ HINs are tdTomato$^+$; very few of them are *Htr3a*$^+$ (CGE marker), suggesting that the MGE/POA is the main origin of these HINs ((*Figure 5I–M*, *Figure 5—figure supplement 1*). In the *Maf* cDKO there was ectopic *Pnoc* expression in the neocortex (highest in the dorsomedial regions abutting the hippocampus), as well as increased *Pnoc*$^+$ cells in the hippocampus, consistent with an increased density of HINs (*Figure 5A–L*) (WT vs cDKO: Neocortex p<0.01; dorsomedial Neocortex p<0.0001; Hippocampus p<0.01; t test). Indeed, *Sst* FISH and tdTomato$^+$ IN quantifications both suggested increased HIN density in the *Maf* cDKO (*Figure 1—figure supplement 7*, *Figure 5—figure supplement 1*).

Together, these data provide evidence that loss of *Maf* and *Mafb* leads to increased production of HINs, which are mainly Sst$^+$ and/or *Pnoc*$^+$. To further test this hypothesis, we studied HIN generation in the cDKO.

## *Maf* and *Mafb* repress the generation of HINs

Since *Maf* cDKOs have increased density of *Pnoc*, a gene mainly expressed in the HINs, we postulated that *Maf* and *Mafb* together repress HIN generation. To validate the *Nkx2-1-Cre* tdTomato findings (*Pai et al., 2019*; *Figure 5—figure supplement 1*), we quantified CINs and HINs at P2 using

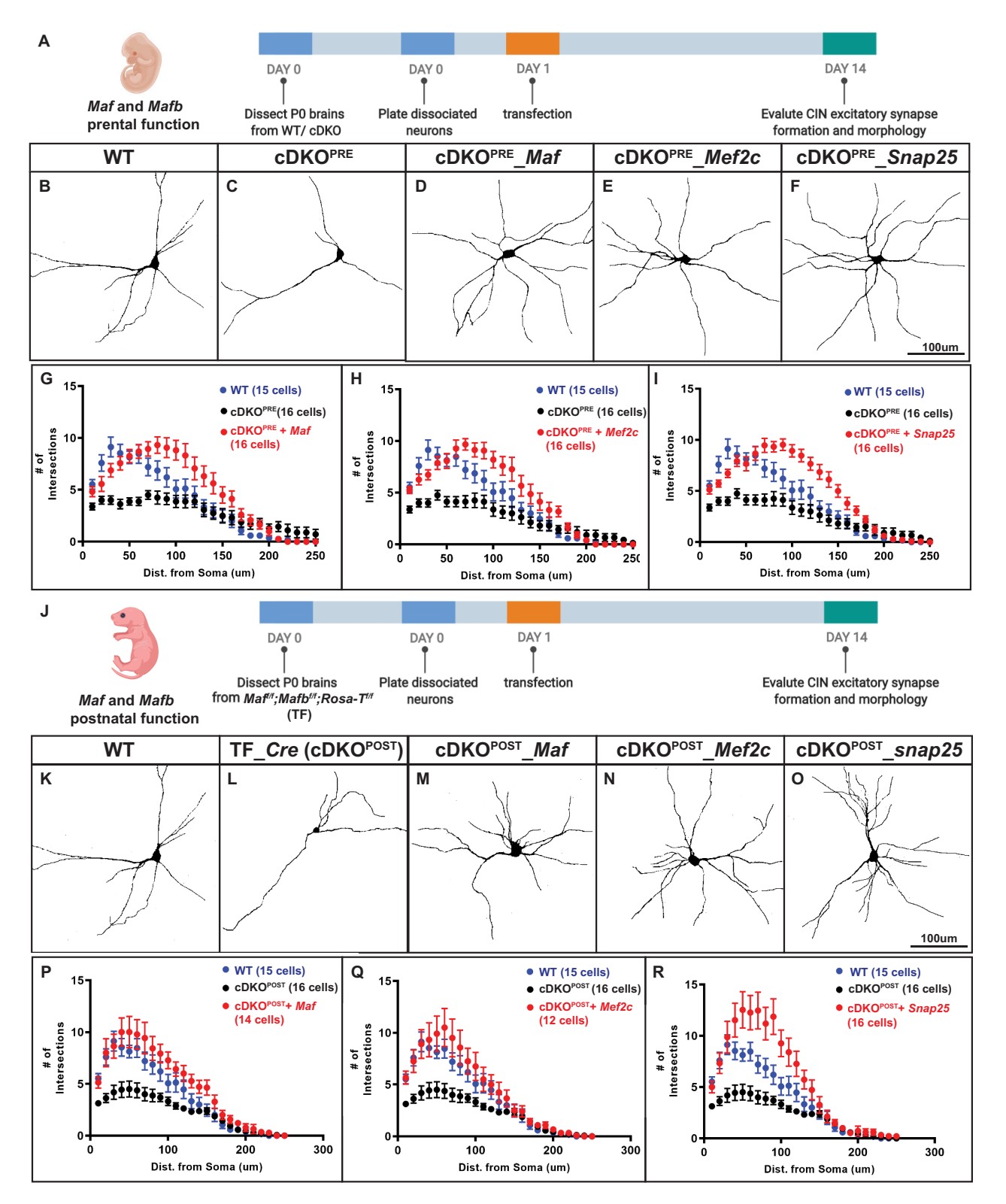

**Figure 4.** *Mef2c* and *Snap25* are required downstream of *Mafb* and *Maf* to promote neurite outgrowth. **A**) Schema showing the in vitro culture assay testing *Maf* and *Mafb*'s prenatal function on neurite outgrowth. P0 WT and cDKO neocortical tissues were dissociated (INs were labeled with *Nkx2.1-cre*-driven tdTomato) and diluted 5-fold with reporter negative dissociated P0 WT neocortical tissues before plating on culture slides. (**B–C**) Representative traces of cultured WT and cDKO INs. (**D–F**) Representative traces of cultured cDKO INs transfected with *Maf, Mef2c* or *Snap25-*

*Figure 4 continued on next page*

*Figure 4 continued*

expressing plasmids; each showed a rescue phenotype. (**G–I**) Sholl analysis showing the neurite complexity and the rescue effects of *Maf* (**G**), *Mef2c* (**H**) and *Snap25* (**I**). (**J**) Schema showing the in vitro culture assay testing *Maf* and *Mafb*'s postnatal function on neurite outgrowth. (**K–O**) Representative traces of cultured INs using *Maf*^f/f^;*Mafb*^f/f^; RosaT^+^ P0 neocortices to look at the effect of postnatal deletion of *Maf* and *Mafb* on neurite outgrowth and if *Maf*, *Mef2c* and *Snap25* expression showed rescue effects. Thus, postnatal loss of *Maf* and *Mafb* (cDKO^POST^) leads to defects in neurite outgrowth (**K–L**). (**P–R**) Sholl analysis showing the neurite complexity and the rescue effects of *Maf* (**P**), *Mef2c* (**Q**) and *Snap25* (**R**) on cDKO^POST^ INs. Scale bar in (**F**) and (**O**) = 100 um. N = 3–4 per groups. The quantity of cells used for analysis was included in (**G–I**) and (**P–R**).

The online version of this article includes the following figure supplement(s) for figure 4:

**Figure supplement 1.** Individual CINs from transplanted WT and Maf cDKO MGE cells grown in vivo for 30 days.

**Figure supplement 2.** *Maf* cDKO^Sst-IRES-Cre^ showed defects in neurite outgrowth We used in vitro culture assay to test *Mafb* and *c-Maf*'s post-mitotic function on neurite outgrowth.

*Lhx6*^+^ FISH. *Lhx6* marks MGE-derived INs (*Cobos et al., 2006*; *Liodis et al., 2007*). *Lhx6*^+^ CIN density was decreased in *Maf* cDKOs as previously reported (*Pai et al., 2019*; *Figure 6B*). Consistent with our hypothesis, there was an increased density of *Lhx6*^+^ HINs across all hippocampal regions in the *Maf* cDKO (p<0.05; Welch's t test) (*Figure 6C*). Furthermore, there was an increased density of *Sst*^+^ HINs cell density in all hippocampal regions (CA1:~1.5 fold; p<0.001, CA3 and DG:~2 fold; p<0.0001) (*Figure 1—figure supplement 7*).

Next, to test if *Maf* cDKOs generate more HINs at specific developmental stages, we conducted EdU/BrdU pulse-chase experiments. Pregnant mice were injected with BrdU at E12.5, and subsequently with EdU at E15.5. The animals were harvested at P2 to study co-expression of *Lhx6* with BrdU and EdU (*Figure 6A*). We compared the generation of CINs and HINs.

At E12.5 (early-born; *Lhx6*^+^;BrdU^+^), we observed ~2 fold increase in HIN generation in the *Maf* cDKO (p<0.01; t test). The increase was similar across all hippocampal subregions (*Figure 6D, E, H and I*) (CA1: p=0.02; CA3: p<0.001; DG: p<0.01; t test). In contrast, *Maf* cDKOs had a trend toward decreased CIN generation at E12.5 (*Figure 6—figure supplement 1*). At E15.5 (late-born; *Lhx6*^+^; EdU^+^), we observed ~2.5 fold increase in HIN production (p<0.0001; t test). Similarly, the increase was across all hippocampal subregions (*Figure 6F, G, J and K*) (CA1: p<0.001; CA3: p<0.05; DG: p<0.01; t test). We further quantified the ratio of HIN density over CIN density by age and by genotypes. In WTs, the MGE tends to generate HINs/CINs in a 1:2 ratio at both E12.5 and E15.5. In cDKOs, the MGE overproduces HINs, especially at E12.5 (*Figure 6—figure supplement 1*). Together, these data support the hypothesis that loss of *Maf* and *Mafb* leads to HIN overproduction perhaps at the expense of generating CINs.

## Discussion

*Maf* (*c-Maf*) and *Mafb* are co-expressed in the MGE and its lineages. Previously, to elucidate how *Maf* and *Mafb* regulated CIN and HIN development, we compared the transcriptomes of WT and *Maf* cDKO using bulk RNA-seq from E13.5 MGEs – this approach failed to identify differentially expressed genes (data not shown) probably because of the heterogeneity of the tissue. Here, we applied scRNA-seq to compare the transcriptomes of P0 WT and *Maf* cDKO neocortex and hippocampus. Through this approach, we successfully identified genes that were differentially expressed in pallial INs. Our experimental analyses provided evidence that some of the DEX genes contribute to *Maf* cDKO phenotypes: (1) reduced *Mef2c* expression contributed to overproduction of SST^+^ CINs and under production of PV^+^ CINs (*Figure 1* and *Figure 2*); (2) reduced *Snap25* and *Mef2c* contribute to defects in IN morphology (*Figure 4*); (3) reduced *Cxcr4* expression led to defects in IN migration and lamination (*Figure 1—figure supplement 6*). We also discovered that the *Maf* cDKO MGE preferentially generates HINs over CINs (*Figure 5* and *Figure 6*). To our knowledge, this may be the first identification of genes that control the differential production of HINs and CINs.

### Identification of genes that are candidate markers of immature PV INs

Our scRNA-seq analysis not only identified DEX genes between WT and cDKOs, but also provided evidence for molecular markers of immature pallial INs, especially for the PV^+^ subtype, which has been elusive to identify before adolescence. Comparison of *Sst*^+^ and *Sst*^-^ P2 INs identified sets of genes that were differentially expressed between the two groups, including *Arl4d*, *distal-less*

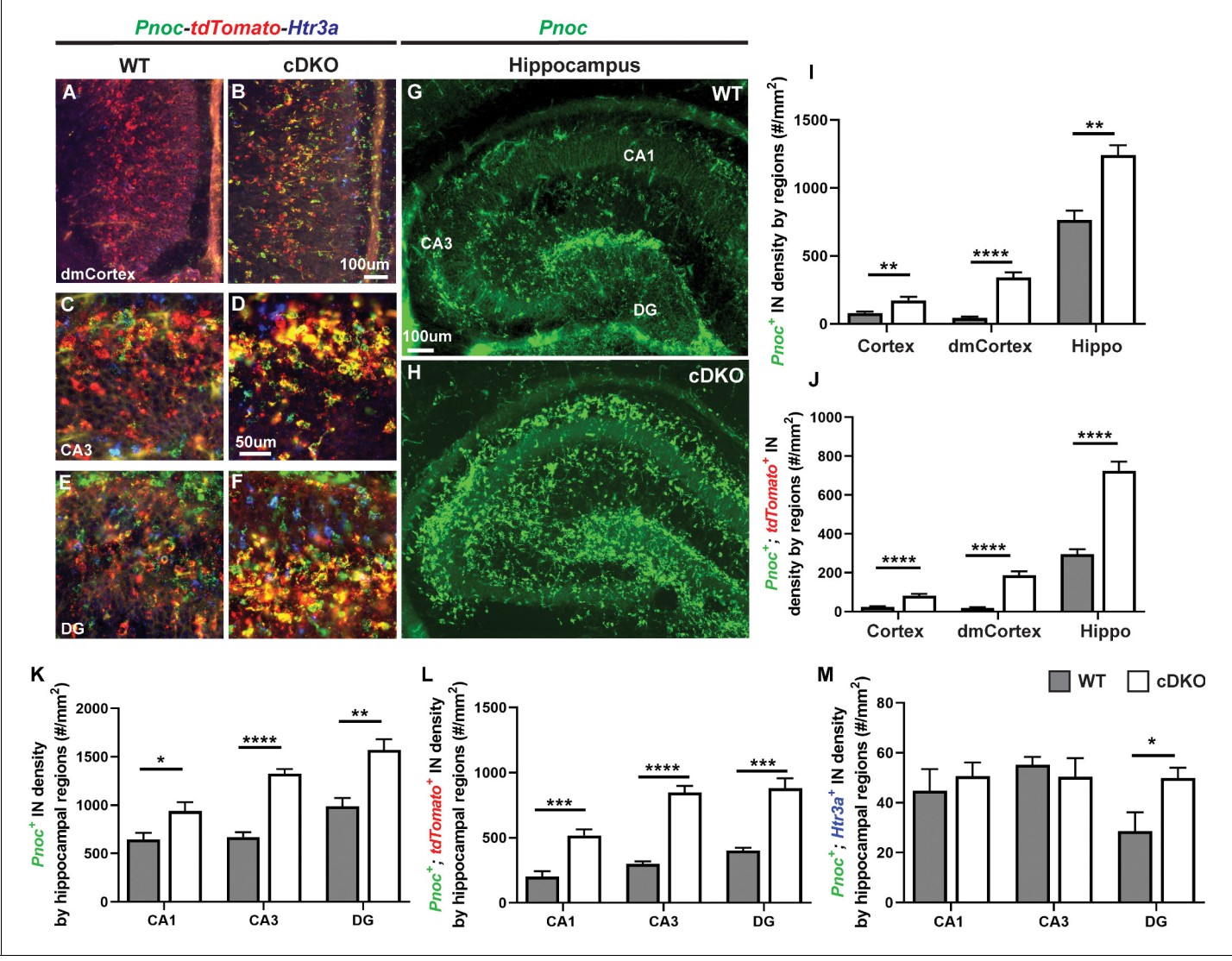

**Figure 5.** *Maf* cDKOs have preferential production of HINs over CINs that are *Pnoc*+. (A–F) Multicolor FISH of *tdTomato*, *Pnoc* and *Htr3a* on P2 WT and cDKO neocortex (A–B) and hippocampus (C–F). (G–H) *Pnoc* expression in the WT and cDKO hippocampus. (I) Quantification of the *Pnoc*+ INs in the neocortex, dorsal-medial cortex (dmCortex) and hippocampus. (J) Quantification of the *Pnoc*+; *tdTomato*+ INs in the neocortex, dmCortex and hippocampus. (K) Quantification of the *Pnoc*+ HINs by regions. (L) Quantification of the *Pnoc*+; *tdTomato*+ HINs by regions. (M) Quantification of the *Pnoc*+; *Htr3a*+ HINs by regions. N = 3–4 per group and multiple sections were quantified per animal. Scale bar in (B) and (G) = 100 um and in (D) = 50 um. *p<0.05, **p<0.01, ***p<0.001, ****p<0.0001. Welch's t test.

The online version of this article includes the following figure supplement(s) for figure 5:

**Figure supplement 1.** Quantification of *Pnoc*+ INs by region and by IN subgroups/Quantification of tdTomato+ HINs by hippocampal region.

homeobox 5 (*Dlx5*), *Elmo1*, Erb-B2 receptor tyrosine kinase 4 (*Erbb4*), *Igfbp4*, *Mef2c*, *Sp9* and *Tcf12* (**Figure 3**). Because *Sst-IRES-Cre* fate mapping into adulthood shows that *Sst-IRES-Cre* expression marks almost all SST+ INs and <10% of PV INs (**Pai et al., 2019**), we hypothesize that P2 *Sst*- INs primarily represent neurons that will become PV+ (**del Rio et al., 1994**). We used multicolor FISH to compare the expression of genes in P2 *Nkx2.1*-lineage *Sst*+ and *Sst*- INs. We validated that *Elmo1*, *Igfbp4*, *Mef2c* and *Sp9* may be candidate markers for immature PV+ INs (**Figure 3**), especially for immature HINs that likely become PV+. This is particularly important and useful for future research that aims to identify different molecular mechanisms involved in PV versus SST HIN development.

Of note, here we report that ~22% of the *Nkx2.1-Cre* lineage CINs are *Sp9* positive at P2. To our knowledge, this is the first report to show *Sp9* expression in CINs in the *Nkx2.1-Cre* lineage. We

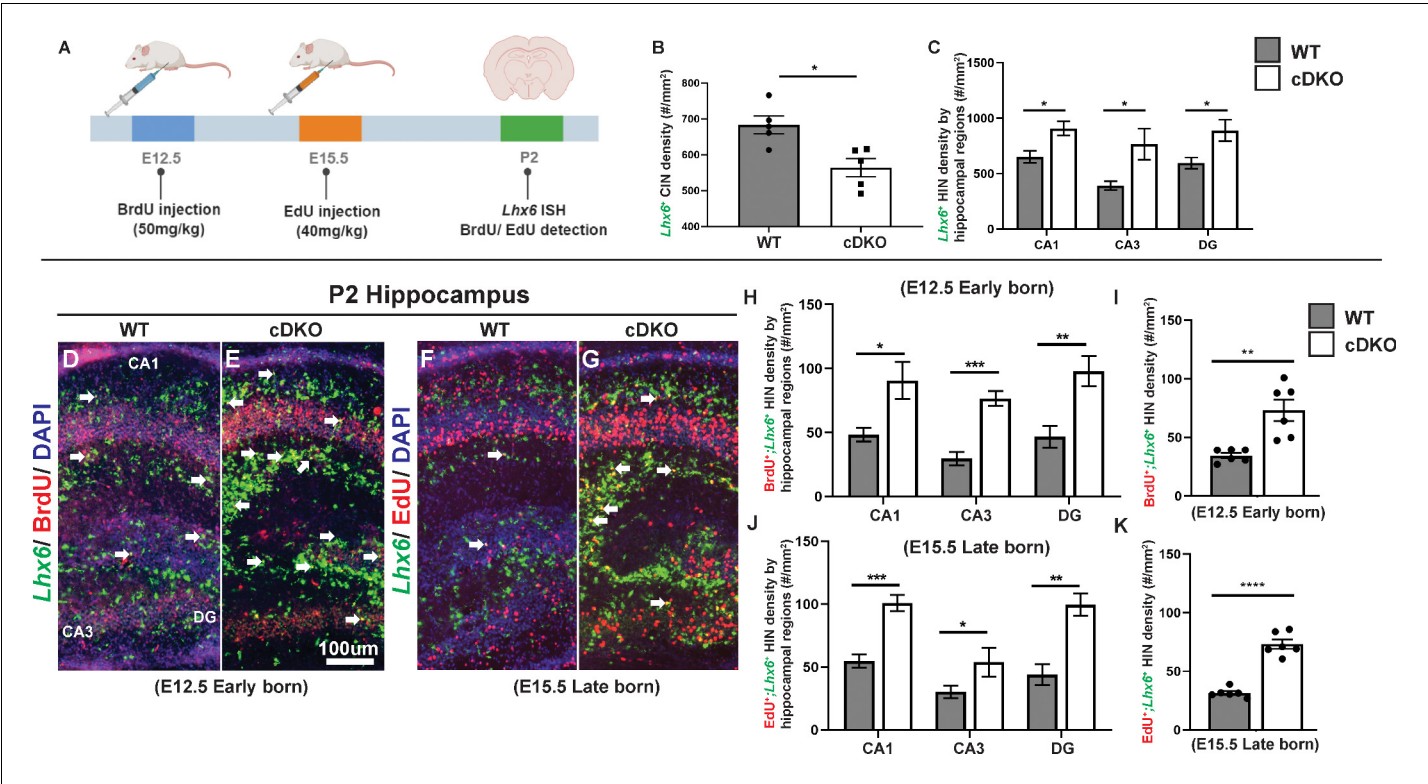

**Figure 6.** *Maf* cDKOs overproduce INs that will become HINs. (**A**) Schema depicting the BrdU/EdU pulse-chase experiment. Briefly, BrdU was injected into a pregnant mouse at E12.5 and EdU was injected at E15.5. Neonatal pups were harvested at P2 analyzed for *Lhx6*/BrdU, *Lhx6*/EdU co-expression. (**B**) Quantification of *Lhx6*⁺ IN cell density in the P2 neocortex. (**C**) Quantification of *Lhx6*⁺ IN cell density in the P2 hippocampus by region. (**D–G**) *Lhx6* FISH with BrdU (D-E, early-born) or EdU (F-G, late-born) co-labeling in P2 WT and cDKO hippocampus. (**H–I**) Quantification of the *Lhx6*⁺; BrdU⁺ IN cell density in the hippocampus by region (**H**) and in total (**I**). (**J–K**) Quantification of the *Lhx6*⁺; EdU⁺ IN cell density in the hippocampus by region (**J**) and in total (**K**). N = 3 per group and multiple sections were quantified per animal. Scale bar in (**E**) = 100 um. Welch's t test.

The online version of this article includes the following figure supplement(s) for figure 6:

**Figure supplement 1.** Analysis of early-born and late-born MGE-derived CINs in WT and *Maf* cDKO.

agree that 22% may be an underestimate because of several reasons: (1) when we looked at the *Sp9* expression in the hippocampus,~60% of the PV candidate INs are *Sp9* positive; (2) the robustness of the *Sp9* RNAscope probe seemed to be stronger in the hippocampus region; (3) the robustness and efficiency of the *Sp9* RNAscope probe is not as strong as the traditional DIG-labeled *Sp9* probe in the neocortex as reported by the Allen Brain Institute. Additionally, recent studies have emerged to study the role of *Sp9* in IN development (*Liu et al., 2019*; *Tao et al., 2019*). Liu et al reported around 50% PV⁺ *Nkx2.1-Cre* lineage CIN loss in adulthood when comparing *Nkx2.1-Cre*-generated *Sp9* conditional heterozygous and homozygous (cHet and cKO) mutants. However, it may be questionable to conclude that 50% of the PV CINs are *Sp9*⁺ due to the fact that *Sp9* deletion also leads to 50% total loss of *Nkx2.1-Cre* lineage CINs (*Liu et al., 2019*). While more studies are needed to validate the roles of these candidate PV IN marker genes in PV IN development and maturation, recent reports on *Sp9* and *Mef2c* have provided a genetic foundation for our discovery of early candidate PV IN markers (*Liu et al., 2019*; *Mayer et al., 2018*; *Tao et al., 2019*).

### *Mef2c* expression driven by *Maf* and *Mafb* promotes PV IN production and morphological maturation

*Mef2c* is a high confidence risk gene for ASD, intellectual disability(ID) and schizophrenia (*Tu et al., 2017*). Notably, *Mef2c* has roles in promoting cortical excitatory neuron synapse formation and morphology maturation (*Harrington et al., 2016*). Furthermore, there is evidence that *Mef2c* marks PV IN precursors, and is required for PV IN development (*Mayer et al., 2018*).

*Mef2c* is one of the most down-regulated genes in *Maf* cDKO MGE derived INs, which we confirmed using immunofluorescence analysis in the P2 *Maf* cDKO neocortex (*Figure 1*, *Figure 1—figure supplement 5*). *Mef2c* rescued the IN neurite outgrowth defect of both prenatal and postnatal *Maf* cDKO INs (*Figure 4*). Thus, *Mef2c* promotes IN morphological maturation, similar to its functions in excitatory neurons (*Harrington et al., 2016*).

Deletion of *Mef2c* using *Dlx5/6-Cre* led to a preferential loss of PV CINs, similar to the *Maf* cDKO phenotype (*Mayer et al., 2018*; *Pai et al., 2019*). Here we found that transduction of *Mef2c* into *Maf* cDKO MGE cells largely rescued their deficit in generating PV CINs (*Figure 2*).

The *Maf* cDKO MGE generates ~2 fold increase in SST$^+$ CINs born at E12.5 and at E15.5, probably at the expense of generating PV$^+$ CINs (*Figure 1*; *Pai et al., 2019*; *Scheme 1*). Thus, in the absence of the *Maf* genes, the MGE has a persistent defect in generating the proper balance of SST/PV INs. We propose a transcriptional pathway in which *Maf* and/or *Mafb* drive *Mef2c* expression in MGE progenitors to promote PV IN production (*Figure 2—figure supplement 1*). This pathway may operate in conjunction with signaling hubs such as MTOR/MAPK/RYK, as perturbation in these pathways also lead to altered PV/SST expression (*Malik et al., 2019*; *McKenzie et al., 2019*; *Angara et al., 2020*). Thus, cellular signaling as well as the *Mafs* and *Mef2c* are critical components of the program driving PV IN development.

*Maf*, *Mafb* and *Mef2c* expression persist as INs migrate into and mature in the pallium. Neonatal deletion of *Maf* and *Mafb* led to reduced neurite outgrowth, a defect rescued by *Mef2c* transduction (*Figure 4*). Thus, our proposed *Maf/Mafb→ Mef2c* transcription pathway persists as INs mature postnatally. Future work should explore whether adult *Maf/Mafb* expression continues to control *Mef2c* and IN function (*Figure 2—figure supplement 1*. Model).

## Mechanisms that underlie PV vs SST IN neurogenesis

While our results support that *Maf* and *Mafb* have important roles in regulating the balance of PV vs. SST IN neurogenesis in the SVZ of MGE (this report and *Pai et al., 2019*), other studies suggest a different hypothesis in which PV INs are primarily generated by the MGE SVZ progenitors, while SST INs are primarily generated by the MGE VZ progenitors (*Glickstein et al., 2007*; *Lodato et al., 2011*; *Petros et al., 2015*). In particular, Petros et al showed that more SST INs are produced when MGE progenitors stay in the VZ (basal progenitor), while more PV INs are produced when MGE progenitors stay longer in the SVZ phase (intermediate progenitor). While the experimental design and results are compelling, the study is based on the hypothesis that MGE-derived IN fates are determined by the progenitor cell cycle stage. Our data showed that *Maf* and *Mafb* expression starts in the MGE SVZ. Deletion of *Maf* and *Mafb* leads to overproduction of SST INs at the expense of PV INs, but not a total loss of PV INs. Therefore, we propose the hypothesis that MGE progenitors have the potential to generate different types of INs through distinct transcriptomic regulation. Future studies are certainly needed to better understand these two hypotheses.

## *Snap25* expression driven by *Maf* and *Mafb* promotes MGE-derived IN morphological maturation

*Snap25* is part of the neuronal SNAP receptor (SNARE) complex which regulates vesicle fusion, receptor trafficking and neurite branching (*Delgado-Martínez et al., 2007*; *Sudhof, 2004*). *Snap25* loss of function studies in hippocampal excitatory neurons causes defects in neurite outgrowth and neuronal survival (*Delgado-Martínez et al., 2007*).

*Snap25* expression was decreased in the *Maf* cDKO INs (*Figure 1—figure supplement 5*). Expressing *Snap25* in prenatal and postnatal *Maf* cDKO INs, rescued their neurite outgrowth defect, similar to what was observed with *Mef2c* (*Figure 4*). Thus, *Maf* and *Mafb* driven expression of both *Snap25* and *Mef2c* are critical aspects of how the *Maf* genes regulate IN development. Since each gene could rescue these phenotypes, it is possible that they may regulate each other.

## *Maf* and *Mafb* repress HIN production and thereby control the balance of CIN and HIN numbers

*Maf* cDKOs had increased numbers of HINs (*Figure 5*). BrdU and EdU birth-dating provide evidence that the cDKO generates more HIN at both early and late stages of MGE neurogenesis (*Figure 6*).

The molecular program that distinguishes CINs from HINs generated by the MGE remains largely unknown. Herein via scRNA-seq, we identified that *Pnoc*, a gene encoding prepronociceptin, may be a unique marker that distinguishes HINs from CINs. *Pnoc* was previously described as an IN marker (*Russ et al., 2015*; *Spiegel et al., 2014*). In a more recent scRNA-seq analysis from hippocampal tissue, *Pnoc* expression was found enriched in O-LM HINs (*Harris et al., 2018*). Our ISH analysis from P2 WTs demonstrated that *Pnoc* is widely and robustly expressed in HINs but not CINs (*Figure 5*). Furthermore, *Pnoc* expression is strongly upregulated in the hippocampus in the cDKO (*Figure 5*), consistent with evidence for increased HIN generation.

Altogether, our data suggests that *Maf* and *Mafb* control the CIN/HIN ratio by putting a brake on HIN generation. As a result, the cDKOs have a disproportionate number of HINs. Thus, in addition to controlling the SST/PV IN subtype fate decision, *Maf* and *Mafb* also control CIN/HIN regional fate determination.

## Decreased *Cxcr4* expression in the *maf* cDKO may alter IN laminar positioning

Our scRNA-seq data suggested that *Cxcr4* expression was reduced in neonatal *Maf* cDKO MGE INs (*Figure 1—figure supplement 5*). Indeed, FISH at P2 showed clear reduction of *Cxcr4* expression in *Maf* cDKO, especially in the hippocampus (CA1, CA3 and DG p=0.0159; Neocortex p<0.0001; t test) (*Figure 1—figure supplement 6*). *Cxcr4* expression reduction starts as early as E15.5 (p=0.012; t test) (*Figure 1—figure supplement 6*). These provide evidence for a potential mechanism underlying premature IN entrance into the cortical plate, that we reported previously (*Pai et al., 2019*).

Note that *Cxcr4* null mice did not show IN over-migration or over-production in the hippocampus (*Sánchez-Alcañiz et al., 2011*; *Stumm et al., 2003*; *Wang et al., 2011*; *Li et al., 2008*). While lacking *Cxcr4* in the cDKO would lead to disruption in the IN chemotactic migration, reduced *Cxcr4* alone probably does not explain why *Maf* cDKOs have more HINs.

## Potential mechanisms that lead to an increased HIN/CIN ratio in the *maf* cDKO

DE analysis suggested that *Nrp1* is upregulated in the cDKO (*Figure 1—figure supplement 5*). FISH showed higher *Nrp1* expression in the P2 cDKO hippocampus (*Figure 1—figure supplement 7*). Semaphorin binding to the *Nrp1* receptor is implicated in MGE-derived IN migration away from the striatum (*Marín et al., 2001*). Semaphorin/neuropilin signaling is also implicated the generation and migration of CINs (*Andrews et al., 2017*; *Tamamaki et al., 2003*). However, we are unaware of a study that focused on *Nrp1*'s role in promoting the production/migration of HINs vs CINs.

We hypothesized that the increase in *Nrp1* expression might affect CIN/HIN migration or regional fate in the cDKO. We tested this idea by introducing *Nrp1* expression into MGE WT progenitors at E12.5. We transplanted MGE immature INs carrying higher expression of *Nrp1* into the P1 neocortex, and surveyed their final cortical destination at P21. The majority of these modified INs appear to remain as an aggregate at the transplantation site. Very few of them migrated within the neocortex or to the hippocampus (data not shown). These results suggest that an increase in *Nrp1* expression alone is not sufficient to promote the increased HINs/CIN ratio.

Interestingly, *Nrp1* is not only a receptor for semaphorins, but also for vascular endothelial growth factor (VEGF). There is evidence that VEGF is expressed in the developing cortex and may regulate IN migration (*Barber et al., 2018*). Mackenzie and Ruhrberg et al. extensively reviewed the functions of VEGF in the nervous system (*Mackenzie and Ruhrberg, 2012*), which includes regulating neurogenesis (*Jin et al., 2002*), neuronal migration (*Schwarz et al., 2004*) and axonal guidance (*Erskine et al., 2011*). In the developing mouse brainstem, VEGF guides somata migration of NRP1-expressing facial motor neurons (*Schwarz et al., 2004*). In the visual system, VEGF serves as a chemoattractant to NRP1-expressing retinal ganglionic cells to promote contralateral axonal extension (*Erskine et al., 2011*). Another study further suggested that VEGF-NRP1 binding works in conjunction with CXCR4-SDF1 to promote breast cancer cell invasion/metastasis (*Bachelder et al., 2002*).

Thus, while the change of *Cxcr4* and *Nrp1* expression in the *Maf* cDKO alone does not provide a clear mechanism for why the mutant generates more HINs, we propose that interactions between NRP1 and CXCR4 can guide IN migration, and that in this context, increases in NPR1 signaling may guide INs to the hippocampus.

## *Maf* cDKOs exhibit seizures - disease implications of *Maf* and *Mafb* dysfunction

EEG recordings from freely behaving mice showed that *Maf* cDKOs exhibit frequent spike-and-wave-discharges (*Figure 1—figure supplement 1*) similar to those that have been previously described in mouse models of genetic *absence* type epilepsy (*Crunelli and Leresche, 2002*; *Paz et al., 2011*; *Sorokin et al., 2016*; *Sorokin et al., 2017*). The cellular and molecular mechanisms underlying this are undoubtedly multifaceted, and include the overall reductions in MGE-derived CINs and HINs in the adult *Maf* cDKO mutant (*Pai et al., 2019*). Furthermore, we identified several genes regulated by *Maf* and *Mafb*, including *Mef2c*, *Snap25* and *Npy*, mutations of which could lead to seizures (*Colmers and El Bahh, 2003*; *Harrington et al., 2016*; *Houenou et al., 2017*). Moreover, human patients with *Maf* mutations exhibit seizures and ID (*Niceta et al., 2015*).

In summary, we identified genetic targets (e.g. *Mef2c*, *Snap25*, *Cxcr4*) that are downstream of *Maf* and *Mafb*. While *Maf* and *Mafb* themselves are not high confidence neuropsychiatric disease genes, they are key regulators of many disease genes that are responsible for balanced MGE SST and PV IN production (*Mef2c*), IN morphological maturation (*Mef2c*, *Snap25*), and balanced MGE CIN and HIN production (*Cxcr4*, *Nrp1*).

## Materials and methods

### Key resources table

| Reagent type (species) or resource | Designation | Source or reference | Identifiers | Additional information |
|---|---|---|---|---|
| Genetic reagent (*M. musculus*) | *Mafb*^f/f | *Yu et al., 2013* | | Generous gift from Dr. Lisa Goodrich |
| Genetic reagent (*M. musculus*) | *Maf*^f/f | *Wende et al., 2012* | | Generous gift from Dr. Carmen Birchmeier |
| Genetic reagent (*M. musculus*) | *Nkx2.1-Cre* | *Xu et al., 2008* | Jax labs #008661 | |
| Genetic reagent (*M. musculus*) | *Sst-IRES-Cre* | *Taniguchi et al., 2011* | Jax labs #013044 | |
| Genetic reagent (*M. musculus*) | *Ai14 Rosa26-td Tomato (RosaT)* | *Madisen et al., 2010* | Jax labs #007908 | |
| Antibody | Rabbit anti-PARVALBUMIN (polyclonal) | Swant | PV27 RRID:AB_2631173 | 1:400 dilution; no need for antigen retrieval |
| Antibody | Rat anti-SOMATOSTATIN (monoclonal; clone YC7) | Millipore Sigma | MAB354 RRID:AB_2255365 | 1:200 dilution; works well in low fixed tissue |
| Antibody | Rabbit anti-MEF2C (polyclonal) | Proteintech | 10056–1-AP RRID:AB_513447 | 1:500 dilution; no need for antigen retrieval |
| Antibody | Rat anti-BrdU antibody (monoclonal [BU1/75(ICR1)] | Abcam | Ab6326 RRID:AB_305426 | 1:500 dilution; BrdU was stained after FISH procedure (proteinase K-based antigen retrieval) |
| Antibody | Sheep anti-DIG-POD antibody (polyclonal) | Millipore Sigma | 11207733910 RRID:AB_514500 | 1:1000 dilution |
| Antibody | Sheep anti-Fluorescein-POD antibody (polyclonal) | Millipore Sigma | 11426346910 RRID:AB_840257 | 1:1000 dilution |
| Antibody | Chicken anti-GFP (polyclonal) | Aves | GFP-1020 RRID:AB_100000240 | 1:1000 dilution |

*Continued on next page*

*Continued*

| Reagent type (species) or resource | Designation | Source or reference | Identifiers | Additional information |
|---|---|---|---|---|
| Commercial assay or kit | Click-iT EdU kit | Life Technologies | Cat #C10340 | Follow the manufacturer's instruction |
| Commercial assay or kit | RNAscope multiplex fluorescent kit (version 2) | ACD Bio | Cat #323120 | Follow the manufacturer's instruction |
| Commercial assay or kit | Mm-*Igfbp4* | ACD Bio | Cat #425711 | Follow the manufacturer's instruction and used it with the RNAscope multiplex fluorescent kit (version 2) |
| Commercial assay or kit | Mm-*Sst* | ACD Bio | Cat #404631 | |
| Commercial assay or kit | Mm-*Sp9* | ACD Bio | Cat #564531 | |
| Commercial assay or kit | Mm-*Mef2c* | ACD Bio | Cat #421011 | |
| Commercial assay or kit | Mm-*Tcf12* | ACD Bio | Cat #504861 | |
| Commercial assay or kit | Mm-*tdTomato* | ACD Bio | Cat #317041 | |
| Commercial assay or kit | Mm-*Pnoc* | ACD Bio | Cat #437881 | |

## Animals

All procedures and animal care were approved and performed in accordance with the University of California San Francisco Laboratory Animal Research Center (LARC) guidelines. All mice strains have been previously published: Ai14 Rosa 26-tdTomato *Cre*-reporter (**Madisen et al., 2010**), *Nkx2.1-Cre* (**Xu et al., 2008**), *Sst-IRES-Cre* (**Taniguchi et al., 2011**), *Mafb* flox (**Yu et al., 2013**) and *Maf* flox (**Wende et al., 2012**). Mice were back-crossed onto a CD-1 background before analyses. For timed pregnancies, noon on the day of the vaginal plug was counted as embryonic day 0.5.

## EdU/BrdU injections and analysis

For pulse-chase experiments, pregnant mice were pulsed with 5-Bromo-2′-deoxyuridine (BrdU, Sigma B5002, prepared at 20 mg/ml) at E12.5 at a dose of 50 mg BrdU/kg body weight and 5-Ethynyl-2′-Deoxyuridine (EdU, Thermo fisher Scientific E10187, prepared at 10 mg/ml) at E15.5 at a dose of 40 mg EdU/kg body weight. Progeny were sacrificed at P2. BrdU$^+$ cells were visualized using anti-BrdU antibody (Abcam, #ab6326). EdU$^+$ cells were visualized using standard procedures from the Clik-iT EdU plus kit (Thermo Fisher Scientific C10340). Of note, BrdU staining was done following *Lhx6* FISH. Since the tissues were processed mainly for FISH, tissue permeabilization was done using proteinase K for 30 mins. There was no hydrogen chloride (HCl) treatment for the tissue. This step likely decreased the sensitivity of the BrdU antibody. Although the BrdU antibody we used has been reported previously for the cross-reactivity with EdU, since we skipped the HCl pre-treatment, we did not observe much cross-reactivity, especially in the *Lhx6*$^+$ IN population.

## Immunofluorescence/Immunohistochemistry

All P2 animals were perfused with 4% PFA, followed by 4% PFA overnight fixation. Tissues were transferred to 30% sucrose the next day for cryoprotection. Brains were embedded with OCT and tissues were sectioned coronally at 25 μm. Sections were all mounted onto glass slides before staining or ISH procedures. To perform immunofluorescence staining, all primary and secondary antibodies were diluted in PBS containing 2.5% BSA and 0.3% Triton X-100. For all experiments, primary antibodies were incubated with tissue overnight, while secondary antibodies were incubated for at least 2 hr in room temperature (RT). Immunofluorescent labeling was performed with the following primary antibodies: rabbit anti-parvalbumin (Swant PV27; 1:400), rat anti-somatostatin (Millipore

MAB354; 1:400), rabbit anti-MEF2C (proteintech 10056–1-AP). The appropriate 488, 594 or 647 Alexa-conjugated secondary antibodies (1:800-1:1000) were from Life Technologies. After the staining, sections were cover slipped with Vectashield containing DAPI (Vector labs; H-1800).

## Fluorescent In situ hybridization (FISH) and standard ISH

To generate riboprobes for target genes, cDNAs were PCR amplified from a homemade mouse cDNA library synthesized from P0 neocortex using Superscript II. The 5' and 3' primers while containing sequences for cDNA synthesis, also introduced ClaI and XbaI restriction enzymes sites. The PCR products and the vector, pSP73 (Promega Cat # P2221), were digested with ClaI and XbaI, and then ligated using T4 ligase. The RNA anti-sense fluorescein-labeled/Digoxigenin-labeled probe was generated by T7 RNA polymerase (Roche) and Fluorescein labeling kit (Roche)/DIG labeling kit (Roche) from a NdeI linearized vector, with the size of the probe usually between ~600–800 bp. Briefly, 40 um sections were digested with proteinase K for 20 mins followed by 4% PFA post-fixation and 3% H2O2 incubation to block endogenous peroxidase. Hybridization with DIG and/or Fluorescein-labeled probes were performed in prehybridization buffer at 65°C overnight. Standard ISH was amplified with BM purple after anti-DIG-AP labeling, while FISH signals were performed following TSA plus kit (Akoya Biosciences) protocol after anti-DIG-POD or anti-Fluorescein-POD labeling. Detailed materials and steps can be found as previously published (*Duan et al., 2018*).

## RNAscope multi-color fluorescent in situ hybridization

Tissues were all sectioned at 25 um and mounted onto microscope slides before the experiment. The day before the experiment, tissues were defrosted and baked at 60°C for an hour for better attachment followed by PFA post-fixation at 4°C overnight. The in situ hybridization was performed following ACD protocol for the multiplex fluorescent kit (version 2). The RNAscope probes were as follows: Mm-Igfbp4 (Cat. 425711), Mm-Sst (Cat. 404631), Mm-Sp9 (Cat. 564531), Mm-Mef2c (Cat. 421011), Mm-Tcf12 (Cat. 504861), tdTomato (Cat. 317041) and Pnoc (Cat. 437881). Slides were mounted with Vectashield mounting media containing DAPI (Cat. H-1800). For the PV marker discovery experiment, we used multiple sections from 2 WT animals for validation and quantification.

## Single cell RNA-seq sample and library preparation

For single cell dissociation, primary neocortex tissues were dissected (1 P0 WT and 1 P0 cDKO; age and gender matched), and incubated with a pre-warmed solution of Papain (BrainBits) prepared according to manufacturer's instructions for 15 min at 37°C. After incubation, tissues were washed with DMEM/10% FBS (Thermo Fisher) solution multiple times, followed by gentle trituration. Samples were diluted to approximately 2,000 cells per microliter. Cell concentration was confirmed using a CountessII automatic hemocytometer before loading onto a 10X genomics single cell preparation platform following manufacturer's instruction to target 10,000 cell recovery per sample. Library preparation was performed by UCSF Institute of Human Genetics Core (IHG), using 10X Chromium Single 3' Reagent Kits Version 2 (10X Genomics). The library length and concentration were quantified using a Bioanalyzer (Agilent) according to manufacturer's protocol. Paired-end sequencing was performed on the Illumina NovaSeq.

## Processing of sequencing reads

Raw base call (BCL) files were converted into FASTQ files and demultiplexed to align cells with their own barcodes using Cell Ranger V.2. Transcriptome alignment (mm10), cell filtering, barcode counting, UMI counting as well as matrix data output for downstream clustering and differential expression analyses were also performed using Cell Ranger V.2 following manufacturer's manual. Up to this point, the pipeline was completely ran by UCSF IHG computational core.

## Cell clustering, *t*-SNE/UMAP visualization and marker-gene identification

A digital gene expression matrix was constructed from the raw sequencing data as described using Cell Ranger. Cells with fewer than 500 UMIs or over 20,000 UMIs were discarded. Downstream analyses were performed with Seurat v.2 and the R package (*Butler et al., 2018*; *Stuart et al., 2018*). First, cells that have genes with very few counts or cells that were with high mitochondrial genes

were filtered out for the downstream analyses. Gene expression value was normalized based on Seurat pipeline suggestion. Briefly, the 'LogNormalize' command normalizes the feature barcode (UMI) expression measurements by the total batch UMI counts; the outputs were multiplied by a scale factor of 10,000 and log-transformed. After normalization, data were scaled based on cell-cell variation regression (we called these numbers normalized expression level) to minimize the unwanted variations likely coming from technical noise or batch effects. The top ~1000 genes with the highest variance were selected to perform principal component analysis (PCA) for dimensionality reduction. 20 PC components were used to perform Louvain clustering with resolution = 1.0 to generate t-SNE plot for cluster visualization. 24 clusters were identified. We then performed FindAll-Marker function from Seurat to identify markers for each cluster. Clusters were assigned to known cell types based upon cluster-specific markers. DE genes between WT and cDKO were performed per cluster basis using MarkerComp function. DE genes with adjusted p-value less than 0.05 and |LogFC| > 0.3 were selected to perform ISH for validation and functional studies. Of note, we also performed unsupervised clustering, marker gene identification, DEX analysis and UMAP visualization using Scanpy with Python (*Wolf et al., 2018*). We found consistent results between the two unbiased analyses.

## MGE transplantation

A detailed protocol for this procedure is available in a methods format (*Vogt et al., 2015*; *Vogt et al., 2017*). We bred male mice that would yield embryos carrying *Ai14* and either WT (*Ai$^{Flox/Flox}$*) or homozygous (*Maf$^{Flox/Flox}$*; *Mafb$^{Flox/Flox}$*; RosaT$^+$) for both of the *Maf* alleles. The embryos were collected at E13.5, dissociated and then transfected either with a *Dlxl12b-Cre* expressing vector or both *Dlxl12b-Cre* expressing and target gene expressing vectors (*Vogt et al., 2015*; *Vogt et al., 2017*) in DMEM supplemented with 10% Fetal Bovine Serum (FBS) at 37°C and at pH ~7.2 for 30 min. The successful delivery of the *Cre* DNA will deletes the *Maf* genes and activates tdTomato expression from the *Ai14* allele. The cells were then washed several times with DMEM/FBS and pelleted. Next, the cells were loaded into the front of a beveled glass needle. P1 WT pups were anesthetized on ice and injected with ~300 nL of cells over 3–5 sites in the right hemisphere. Pups were warmed until able to move and then put back with their mom. They were aged to 30 days and then perfused. Note that we section whole brains of all tissues into 25 um sections, and we used all brain sections for PV and SST staining followed by cell quantification.

## Primary neuronal culture for analysis of dendritic arborization (Sholl) and synapses using neonatal cortex

Primary cortical neuron cultures were prepared as described (*Shepherd et al., 2006*). Briefly, we bred *Maf$^{Flox/Flox}$*, *Mafb$^{Flox/Flox}$* RosaT$^+$ females with *Maf$^{Flox/+}$*, *Mafb$^{Flox/+}$*, *Nkx2-1 Cre$^+$* males to generate P0 *Maf* cKO, *Mafb* cKO and cDKOs. Control P0 animals were generated either through the same crossing or through *Ai14f$^{Flox/Flox}$* females bred with *Nkx2-1 Cre$^+$* males. tdTomato$^+$ P0 pups were pre-screened using fluorescence dissection microscope. We also collected some tdTomato$^-$ P0 pups for cell preparation. Cortical tissues were dissected in cold EBSS, followed by trypsin (Thermo Fisher Scientific 25200056) treatment for 15 min at 37°C. Trypsinization was inhibited using 10% FBS containing DMEM. Cells were washed once with DMEM, then resuspended in 10% FBS containing Neuralbasal-A medium (Thermo Fisher Scientific 12348017) with B27 (Thermo Fisher Scientific 17504044). Cell density was quantified using hemocytometer. tdTomato$^+$ cell preparations were diluted using tdTomato$^-$ P0 cell preparation roughly at a ratio of 1:10. Cells were plated onto poly-D-lysine and laminin coated coverslips (Corning 08-774-385) preloaded in 24-well plates and then cultured in a 37°C incubator for 14 days. Serum free Neuralbasal-A medium with B27 and Glutamax (Thermo Fisher Scientific 35050061) was used to maintain the cell growth. After 14 days in vitro, cell culture medium was removed and replaced with freshly made 4% PFA for 15 min fixation. PFA was washed off several times with 1X PBS followed by a regular immunostaining protocol. For Sholl analysis,≥3 animals and 15–20 neurons were analyzed per genotype. Images were processed and analyzed using FIJI software based on previously described protocol (*Pai et al., 2019*) .

## EEG recordings and analysis

EEG recordings and analysis were done as previously described (*Clemente-Perez et al., 2017*; *Ritter-Makinson et al., 2019*). The devices for EEG recordings in freely behaving mice were all custom made in the Paz lab. Briefly, we designed devices containing multiple screws for acquisition of the EEG signals implanted in primary somatosensory (S1), primary visual (V1), and prefrontal cortices. S1: −0.5 mm posterior from Bregma,±3.25 mm lateral. V1: −2.9 mm posterior from Bregma,±2.5 mm lateral. PFC: 1 mm anterior from Bregma, midline. Mice were allowed to recover for at least 1 week before recording. EEG signals were recorded using RZ5 (TDT) and sampled at 1221 Hz. A video camera that was synchronized to the signal acquisition was used to continuously monitor the animals. Each recording trial lasted 30–180 min. To control for circadian rhythms, we housed our animals using a regular light/dark cycle, and performed recordings between roughly 11:00 AM and 6:00 PM. All the recordings were performed during wakefulness.

## Image acquisition and processing

1. Immunohistochemistry images were taken using 10X or 20X objectives under Nikon Ti microscope with DS-Qi2 color camera together with NIS Elements acquisition software. Image brightness and contrast were adjusted and merged for publication using FIJI software.
2. Sholl analysis and excitatory synapse images were taken using the Nikon Ti inverted fluorescence microscope with CSU-W1 large field of view confocal. Images for Sholl analysis were taken using 40X oil objective. Open source micromanager 2.0 beta was used to acquire images. Brightness/contrast adjustment, z-stack image and binary image processing and Sholl analysis were all conducted using the image calculator and Sholl analysis plugins in FIJI software.

## Schema and Model Illustration

All schemas were drafted using the BioRender online platform.

## Quantification and statistical analyses

All bar graphs were shown as mean ± SEM. All statistical analyses were performed using Graphpad Prism (version 7) and a p value of < 0.05 was considered significant. For RNA expression comparison shown in *Figure 3*, we used python to conduct non-parametric analysis, and a p value of < 0.05 was considered significant. The specific n for each experiment as well as the post-hoc test and corrected p-values, can be found in the Results section, in the Figure legends or in supplementary tables.

For all cell counts performed for immunohistochemistry and in situ hybridization, we used the cell counter plug-in in FIJI software. For P2 hippocampus immunofluorescence and FISH cell counts, we focused on the central part of the hippocampi (exclude very rostral and very caudal region). Note that for P2 HIN *Nrp1* FISH, we excluded the pyramidal cell layer of hippocampi for quantification. The *Nrp1* expression is very high in the hippocampal excitatory neurons (pyramidal layer) and we think that might confound the analysis.

For statistical analyses, we mainly used two methods depending on whether data were parametric or nonparametric in distribution. For parametric data, we utilized a Welch's t test. For data that was normalized (eg. proportions), we used the non-parametric Mann-Whitney test. For MGE transplantation, we used non-parametric *Chi*-square test for comparison.

## Acknowledgements

This work was supported by the following research grants. JLRR is supported by Nina Ireland, NIMH R01 MH081880, NIMH R37/R01 MH049428, NIH/NIDDK P30DK098722 (UCSF NORC); DV is funded by Spectrum Health-MSU Alliance Corporation. ELLP is funded by the UCSF neuroscience graduate program and NIMH R01 MH081880. Jin Chen is funded by the Jane Coffin Childs Memorial Fund for Medical Research and the NIH K99/R00 Pathway to Independence Award (GM134154); SFD is funded by NINDS R01 NS34661 and Simons Foundation (SFARI A133320); MFP and JSC are funded by NIH/NINDS K08NS091537; JTP is supported by NIH/NINDS R01NS096369, Gladstone Institutes, and NSF # 1608236. FSC is supported by the National Science Foundation Graduate Research Fellowship Award (NSF #1144247), NIH F31 NS111819-01A1 and the UCSF neuroscience graduate

program; Jiapei Chen is supported by the UCSF biomedical sciences graduate program. We thank John Askins for assistance with blinded seizure analysis. We thank UCSF Institute of Human Genomics (IHG) for the assistance with 10X single-cell RNA sequencing library preparation and sequencing.

## Additional information

### Competing interests

John LR Rubenstein: is cofounder, stockholder, and currently on the scientific board of Neurona, a company studying the potential therapeutic use of interneuron transplantation. The other authors declare that no competing interests exist.

### Funding

| Funder | Grant reference number | Author |
|--------|------------------------|--------|
| National Institute of Mental Health | MH081880 | Emily Ling-Lin Pai<br>John LR Rubenstein |
| National Institute of Mental Health | MH049428 | John LR Rubenstein |
| National Institute of Diabetes and Digestive and Kidney Diseases | P30DK098722 | Emily Ling-Lin Pai<br>John LR Rubenstein |
| NIH Office of the Director | GM134154 | Jin Chen |
| National Institute of Neurological Disorders and Stroke | NS34661 | Siavash Fazel Darbandi<br>John LR Rubenstein |
| Simons Foundation | SFARI A133320 | Siavash Fazel Darbandi<br>John LR Rubenstein |
| National Institute of Neurological Disorders and Stroke | K08NS091537 | Julia S Chu<br>Mercedes F Paredes |
| Spectrum Health-MSU Alliance Corporation | | Daniel Vogt |
| National Institute of Neurological Disorders and Stroke | R01NS096369 | Frances S Cho<br>Jiapei Chen<br>Jeanne T Paz |
| National Science Foundation | 1608236 | Frances S Cho<br>Jeanne T Paz |
| National Science Foundation | 1144247 | Frances S Cho |
| National Institute of Neurological Disorders and Stroke | F31 NS111819-01A1 | Frances S Cho |
| The Roberta and Oscar Gregory Endowment in Stroke and Brain Research | | Mercedes F Paredes |
| University of California, San Francisco | Neuroscience Graduate Program | Emily Ling-Lin Pai<br>Frances S Cho |
| Jane Coffin Childs Memorial Fund for Medical Research | | Jin Chen |
| Gladstone Institutes | | Jeanne T Paz |
| University of California, San Francisco | Biomedical Sciences Graduate Program | Jiapei Chen |

The funders had no role in study design, data collection and interpretation, or the decision to submit the work for publication.

## Author contributions
Emily Ling-Lin Pai, Conceptualization, Data curation, Formal analysis, Supervision, Validation, Methodology, Writing - original draft, Writing - review and editing; Jin Chen, Data curation, Formal analysis, Validation, Investigation, Methodology, Writing - review and editing; Siavash Fazel Darbandi, Validation, Methodology, Writing - review and editing; Frances S Cho, Formal analysis, Investigation, Methodology, Writing - original draft, Writing - review and editing; Jiapei Chen, Formal analysis, Investigation, Methodology, Writing - review and editing; Susan Lindtner, Investigation, Methodology, conducted experiments, which generated data that we used to address some of referees' questions; Julia S Chu, Supervision, Validation, Methodology, Writing - review and editing; Jeanne T Paz, Data curation, Formal analysis, Funding acquisition, Methodology, Writing - original draft, Writing - review and editing; Daniel Vogt, Conceptualization, Validation, Investigation, Methodology, Writing - review and editing; Mercedes F Paredes, Conceptualization, Resources, Supervision, Investigation, Methodology, Writing - review and editing; John LR Rubenstein, Resources, Supervision, Funding acquisition, Investigation, Writing - original draft, Project administration, Writing - review and editing

## Author ORCIDs
Emily Ling-Lin Pai ⓘ https://orcid.org/0000-0002-6967-5239
John LR Rubenstein ⓘ https://orcid.org/0000-0002-4414-7667

## Ethics
Animal experimentation: All procedures and animal care were approved and performed in accordance with the University of California San Francisco Laboratory Animal Research Center (LARC) guidelines. All animals were handled based on the approved institutional animal care and use committee (IACUC) protocol (AN180174-01B) at the University of California San Francisco.

## Decision letter and Author response
Decision letter https://doi.org/10.7554/eLife.54903.sa1
Author response https://doi.org/10.7554/eLife.54903.sa2

# Additional files
## Supplementary files
• Transparent reporting form

## Data availability
We submitted the original source data that was used for Seurat pipeline analysis to GEO under the accession number GSE144222. Readers can utilize these datasets for reanalysis and new analysis using Seurat pipeline or other customized codes for more data mining.

The following dataset was generated:

| Author(s) | Year | Dataset title | Dataset URL | Database and Identifier |
|---|---|---|---|---|
| Pai ELL, Chen J, Darbandi SF, Cho FS, Lindtner S, Chu JS, Paz JT, Vogt D, Paredes MF, Rubenstein JLR | 2020 | Function of Mafb and c-Maf in MGE-derived interneuron development | https://www.ncbi.nlm.nih.gov/geo/query/acc.cgi?acc=GSE144222 | NCBI Gene Expression Omnibus, GSE144222 |

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
