## [Decision Letter]

**Acceptance summary:**

Through detailed analysis of the *Maf/Mafb* double mutant mice, the manuscript uncovers a set of transcriptionally regulated genes of MGE-derived cortical and hippocampal interneurons. The manuscript combines single cell sequencing and transplantation rescue assay. The results demonstrate arole for these genes in interneuron fate and maturation in part through regulating transcription of neuropsychiatric disease genes.

**Decision letter after peer review:**

Thank you for submitting your article "*Mafb* and *c-Maf* control pallial interneuron fate and maturation through regulation of neuropsychiatric disease genes" for consideration by *eLife*. Your article has been reviewed by three peer reviewers, and the evaluation has been overseen by a Reviewing Editor and Marianne Bronner as the Senior Editor. The following individual involved in review of your submission has agreed to reveal their identity: Stewart Anderson (Reviewer #1).

The reviewers have discussed the reviews with one another and the Reviewing Editor has drafted this decision to help you prepare a revised submission. In recognition of the fact that revisions may take longer than the two months we typically allow, until the research enterprise restarts in full, we will give authors as much time as they need to submit revised manuscripts.

Summary:

In this manuscript, the authors examine the role of *Mafb* and c-*Maf* in the generation of subsets of cortical (CINs) and hippocampal interneurons (HINs). By performing a double conditional knock out using *Nkx2.1-cre*, they then perform single cell RNA-seq analysis on P0 neocortex from wildtype and cDKO animals, revealing upregulation (*Pnoc* and *Sst*) and downregulation (*Mef2c*, *Snap25* and *Nxph1* and *2*) genes. These hits were verified by multi-channel RNAscope. Authors show an increase in somatostatin (SST) CINs and a concomitant decrease in parvalbumin (PV) CINs. This appears to be mediated, at least in part, via *Mef2c* which can specify PV INs. Moreover, while the maturing PV INs have not been well characterized by molecular markers, the authors provide evidence that in addition to *Mef2c*, *Elmo1* and *Igfbp4* are candidate markers for maturing PV INs of the hippocampus. Moreover, *Maf* cDKOs exhibit reduced CINs and increased HINs that express a HIN-specific marker *Pnoc*. This paper follows up a previous one from the same group exploring the roles of *Mafb* and *c-Maf* in cortical interneuron development in mice. The overall experimentation and analyses our excellent, and the impact appropriate. The use of the system to identify candidate genes enriched in immature PV+ cells (well before PV expression) is outstanding. The strengths of this manuscript are: 1) High quality scRNA-seq dataset of P0 neocortical neurons in presence absence of *Mafb*/*c-Maf;* 2) Sequencing results argue in favor of direct cell-autonomous role for both factors – pyramidal neuron and CGE neuron expression profiles largely unchanged; 3) It gives valuable insight into some of the downstream factors that underlie phenotypes described previously; 4) Adds to other evidence that *Mef2c* is an early Pv cell marker; 5) Identifies *Pnoc* as a hippocampal interneuron marker.

Essential revisions:

1) The main weakness is that it serves as something of a companion piece to their previous study, which takes away from much of its novelty. In fact, it is difficult to follow parts of this manuscript without having first read Pai et al., 2019 Cell Reports beforehand. Re-writing the text to reinforce the logical flow of the experiments and perhaps a schematic of the previous findings (something like Figure S2) along with what new has been identified would help the reader appreciate the work described.

2) The use of muti-channel RNAscope to confirm scRNA-seq results needs some improvement. First, it would be helpful – either in the main or in supplemental – to show higher magnification images to clarify what is deemed double-positive. Second, some of the RNAscope images are less convincing, *Tcf12* and *Elmo 1* in particular. New, better images would be needed for these. Finally, Liu et al., 2019 and Tao et al., 2019 both describe the role of *Sp9* in PV cell migration to the cortex and describe dramatic phenotypes relating to the loss of PV cells in the cortex following *Sp9* conditional knockout in the MGE lineage. Given this well described role for *Sp9*, it is surprising that only a subset the tdTomato fate-mapped CINs are *Sp9* (~22% of cortical cells even though PV should encompass ~50-60% of Nkx2-1 lineage) in Figure 3. The authors should address this apparent disparity in the Discussion.

3) Please examine the morphology of SST and/or PV INs in vivo through neuron fills. The in vitro findings are compelling but would be greatly strengthened with by an in vivo analysis. This could also include the SST-cre Maf mutants which, at least the in vitro studies, suggest a role for Mafs in postmitotic INs.

---

## [Author Response]

Essential revisions:1) The main weakness is that it serves as something of a companion piece to their previous study, which takes away from much of its novelty. In fact, it is difficult to follow parts of this manuscript without having first read Pai et al., 2019 Cell Reports beforehand. Re-writing the text to reinforce the logical flow of the experiments and perhaps a schematic of the previous findings (something like Figure S2) along with what new has been identified would help the reader appreciate the work described.

We thank the reviewers for their inputs. In our initial submission, the editor suggested that we should reorganize the manuscript flow. We took the suggestion very seriously to reorganize the paragraphs and figures. These changes were presented in our previous formal submission. In response to the reviewers’ comments, we now schematically summarized the findings from our previous *Cell Reports* paper (Pai et al., 2019) to help readers understand what was previously discovered and to help them understand the questions we address in the current paper. We hope these changes are helpful to the readers (Scheme 1-3 and Figure 3—figure supplement 1).

2) The use of muti-channel RNAscope to confirm scRNA-seq results needs some improvement. First, it would be helpful – either in the main or in supplemental – to show higher magnification images to clarify what is deemed double-positive. Second, some of the RNAscope images are less convincing, Tcf12 and Elmo 1 in particular. New, better images would be needed for these. Finally, Liu et al., 2019 and Tao et al., 2019 both describe the role of Sp9 in PV cell migration to the cortex and describe dramatic phenotypes relating to the loss of PV cells in the cortex following Sp9 conditional knockout in the MGE lineage. Given this well described role for Sp9, it is surprising that only a subset the tdTomato fate-mapped CINs are Sp9 (~22% of cortical cells even though PV should encompass ~50-60% of Nkx2-1 lineage) in Figure 3. The authors should address this apparent disparity in the Discussion.

We have extensively modified Figure 3 based on reviewers’ suggestion. We separated the original Figure 3 to generate (1) a new Figure 3, which highlights the expression of *Mef2c*, *Sp9* and *Igfbp4* in lower and higher magnification; (2) a new Figure 3—figure supplement 2, which highlights the expression of *Tcf12*, *Elmo1* and *Arl4d* in lower and higher magnification; (3) a new Figure 3—figure supplement 3, which provides the higher magnification views of marker expression in the *tdTomato+/Sst-* and *tdTomato+/Sst+* population in HINs.

We agree with the reviewers that some of the RNAscope images are less convincing, *Tcf12* and *Elmo1* in particular. Therefore, we now present higher magnification pictures showing INs that are considered marker positive for each gene (*Igfbp4*, *Sp9*, *Mef2c*, *Tcf12*, *Elmo1* and *Arl4d*) in the neocortex and in the hippocampus (Figure 3, Figure 3—figure supplement 2 and 3). The analysis of the result was conducted at the higher magnification and was easily quantifiable (Figure 3 and Figure 3—figure supplement 2). Furthermore, due to the current shelter-in-place order in San Francisco, all research work has been halted, and our institute does not allow any new experiment to be conducted until further noticed. In addition, the scientist who performed this experiment, prepared the manuscript and drafted this response letter has to leave UCSF by May 25th. Taken together and given these uncertain times, we kindly ask to not have to repeat a new set of RNAscope for *Tcf12* and *Elmo1*.

We understand reviewers’ concern regarding the *Sp9* quantification results. We now have dedicated one paragraph in the Discussion for this:

“Of note, here we report that ~22% of the *Nkx2.1-Cre* lineage CINs are *Sp9* positive at P2. To our knowledge, this is the first report to show *Sp9* expression in CINs in the *Nkx2.1-Cre* lineage. We agree that 22% may be an underestimate because of several reasons: (1) when we looked at the *Sp9* expression in the hippocampus, ~60% of the PV candidate INs are *Sp9* positive; (2) the robustness of the *Sp9* RNAscope probe seemed to be stronger in the hippocampus region; (3) the robustness and efficiency of the *Sp9* RNAscope probe is not as strong as the traditional DIG-labeled *Sp9* probe in the neocortex as reported by the Allen Brain Institute. Additionally, recent studies have emerged to study the role of *Sp9* in IN development (Liu et al., 2019; Tao et al., 2019). Liu et al. reported around 50% PV+ *Nkx2.1-Cre* lineage CIN loss in adulthood when comparing *Nkx2.1-Cre-*generated *Sp9* conditional heterozygous and homozygous (cHet and cKO) mutants. However, it may be questionable to conclude that 50% of the PV CINs are *Sp9+* due to the fact that *Sp9* deletion also leads to 50% total loss of *Nkx2.1-Cre* lineage CINs (Liu et al., 2019). While more studies are needed to validate the roles of these candidate PV IN marker genes in PV IN development and maturation, recent reports on *Sp9* and *Mef2c* have provided a genetic foundation for our discovery of early candidate PV IN markers (Liu et al., 2019; Mayer et al., 2018; Tao et al., 2019).“

3) Please examine the morphology of SST and/or PV INs in vivo through neuron fills. The in vitro findings are compelling but would be greatly strengthened with by an in vivo analysis. This could also include the SST-cre Maf mutants which, at least the in vitro studies, suggest a role for Mafs in postmitotic INs.

Due to the shelter-in-place situation, we are not allowed to conduct any experiment. However, to try to investigate the in vivo and in vitro correlation on *Maf* cDKO IN morphology, we revisited the 2D images from our previous MGE transplantation assay and compiled a collection of WT and *Maf* cDKO IN (these INs developed in a WT environment in vivo). The data suggested that the *Maf* cDKO INs have less neurites, which is consistent with our in vitro finding.

In our previous submission, we did include the *Sst-IRES-Cre Maf* cDKO data (Figure 4—figure supplement 2) to support the hypothesis that *Maf* and *Mafb* are functioning postnatally to mediate the CIN morphogenesis.

In our 2019 *Cell Reports* paper, we did not conduct the in vivo IN reconstruction in adult WT and *Maf* cDKO to support the in vitro morphology finding, which is a drawback of the study. However, we provided several other pieces of evidence in the *Cell Reports* paper to support the close correlation between in vitro and in vivo finding of *Maf* mutants. For example, we discovered that *c-Maf (Maf)* cKO IN has increased excitatory input in vitro. With more *c-Maf (Maf)* cKO IN excitation, we observed decreased cortical excitability in acute brain slices using local field potential (LFP) recording. in vitro cultured *Maf* cDKO INs did not show alterations in excitatory synapse formation. In correspondence to that, we did not observe obvious changes in cortical excitability in the ex vivo LFP recording, We also dissected E13.5 MGE progenitors from the *Maf* cDKO and transplanted these cells into P0 neocortex and allowed them to grow in an in vivo WT environment for 30 days. These transplanted cells demonstrated preferential loss of PV INs as observed in vivo.